# A Survey of Flow Matching in Reinforcement Learning

**Nabuat Zaman Nahim**  *nabuat.nahim@uky.edu*
*Department of Computer Science, University of Kentucky, Lexington, Kentucky 40506, USA*

**Fairoz Nower Khan**  *fairoz.khan@uky.edu*
*Department of Computer Science, University of Kentucky, Lexington, Kentucky 40506, USA*

**Peizhong Ju**  *peizhong.ju@uky.edu*
*Department of Computer Science, University of Kentucky, Lexington, Kentucky 40506, USA*

**Reviewed on OpenReview:** *https://openreview.net/forum?id=P6e5IC4gPe*

## Abstract

Flow Matching (FM) has recently emerged as a principled and efficient generative modeling framework for reinforcement learning (RL), enabling expressive, multimodal policy parameterizations via deterministic probability transport. Compared to diffusion-based policies that rely on stochastic denoising chains, FM uses sampling based on ordinary differential equations (ODEs), with learned velocity fields, which can substantially reduce inference latency and simplify the incorporation of RL objectives. As research in flow-based RL rapidly accelerates across offline continuous control, online fine-tuning, and foundation model alignment, the literature has become highly fragmented. In this survey, we provide a comprehensive taxonomy of flow-matching approaches in reinforcement learning. We organize the literature along two axes: the target distribution being modeled (e.g., action policies, value critics, transition dynamics) and the mechanism of RL signal integration (e.g., energy-weighted regression, flow-based policy gradients, and group relative policy optimization). Furthermore, we survey emerging frontiers such as discrete and non-Euclidean action spaces, provide a systematic comparative analysis against Gaussian and diffusion baselines, and outline critical open problems. Ultimately, this survey serves as a foundational roadmap for the next generation of generative reinforcement learning and alignment.

## 1 Introduction

Flow Matching (FM) and related continuous-time flow models (Lipman et al., 2023; Liu, 2022) have recently emerged as a promising framework for training expressive generative models. Unlike diffusion models (Ho et al., 2020; Song et al., 2021), which learn to reverse stochastic noise processes, flow matching directly learns deterministic vector fields that transport samples from a simple base distribution to a target distribution via ordinary differential equations (ODEs). This formulation converts generative modeling into a supervised regression problem on velocity fields, often simplifying training and enabling efficient sampling with relatively few ODE solver steps. These properties make flow-based models particularly attractive for reinforcement learning (RL), where policies must be both expressive and computationally efficient.

As a result, a rapidly growing literature has begun to explore flow matching based approaches for reinforcement learning from several complementary perspectives. A first line of work studies *flow-based policy learning*, especially in offline RL, where flow policies are trained either by energy or value guided regression toward high return actions (Zhang et al., 2025b; Alles et al., 2025; Zhang et al., 2025a), by combining pretrained flow priors with Q-learning based policy improvement (Park et al., 2025b), or by improving efficiency through single-step completion or structural support constraints (Koirala & Fleming, 2025; Zhang et al., 2026). A second line of work considers *online optimization of flow policies*, including likelihood or reward weighted fine tuning (Fan et al., 2025b; Zhang et al., 2025c), direct policy gradient based optimization through the

flow ODE (McAllister et al., 2025; Pfrommer et al., 2025; Yang et al., 2026), and more recent group-relative optimization methods for large scale generative flow models (Liu et al., 2025a; He et al., 2025; Yu et al., 2026; Chen et al., 2025c). Beyond action policies, recent studies have also used flow matching to model *critics*, *dynamics*, *preferences*, and *multi-agent joint behaviors*, broadening the role of FM in RL beyond policy representation alone (Agrawalla et al., 2025; Kong et al., 2025; Kim et al., 2024; Zhu et al., 2025; Lee et al., 2025a). Together, these works show that flow matching is developing into a general RL toolkit that spans offline policy learning, online fine tuning, alignment, and structured decision making. While these individual contributions are significant, the field currently lacks a systematic overview, making it challenging for researchers to connect core algorithmic principles and form a coherent picture of the landscape.

To address this gap, we present a focused survey of flow matching approaches in reinforcement learning. We synthesize a fragmented literature into a cohesive framework, highlighting how diverse methods are connected through policy optimization via deterministic vector fields and clarifying where and how they fundamentally differ. Our goal is to provide a clear understanding of how flow matching is used in RL, how it differs from diffusion based approaches, and where it provides practical and theoretical advantages for policy learning.

## 1.1 Motivation

A major motivation for integrating generative models into reinforcement learning is the representational limitation of traditional policy parameterizations. In many continuous-control settings, standard RL algorithms parameterize stochastic policies as diagonal Gaussian distributions (Sutton & Barto, 2018). Formally, such policies take the form $\pi_\theta(a|s) = \mathcal{N}(a; \mu_\theta(s), \Sigma_\theta(s))$, where $\mu_\theta(s)$ is the mean action predicted by a neural network and the covariance matrix is constrained to be diagonal, i.e., $\Sigma_\theta(s) = \mathrm{diag}(\sigma^2(s))$.

This diagonal structure implies conditional independence between action dimensions, meaning the policy factorizes into independent one-dimensional Gaussians across each action component. As a result, the policy can only represent a single unimodal distribution centered around $\mu_\theta(s)$.

While this parameterization is computationally convenient and widely used in actor-critic algorithms such as PPO (Schulman et al., 2017) and SAC (Haarnoja et al., 2018), since it is inherently unimodal, it is limited in its ability to represent diverse behaviors. When multiple distinct actions are valid such as navigating either left or right around an obstacle a unimodal policy tends to average across these modes, assigning high probability to intermediate actions that may be unsafe or suboptimal. This phenomenon, often referred to as *mode averaging*, becomes particularly problematic in offline RL and imitation learning settings where datasets may contain multiple valid strategies.

To address this limitation, recent work has explored generative models as expressive policy representations. In particular, diffusion models (Ho et al., 2020; Song et al., 2021) have demonstrated strong empirical performance in offline RL and planning tasks (Janner et al., 2022; Wang et al., 2023). Diffusion-based policies generate actions by iteratively denoising a noisy sample through a sequence of reverse diffusion steps.

However, diffusion models typically require tens or hundreds of iterative denoising steps to produce a single action sample, which introduces significant inference latency. For instance, diffusion-based control policies often require 5–100 neural network evaluations per action (Janner et al., 2022; Wang et al., 2023), compared with a single forward pass for conventional policies. This computational cost can make diffusion models impractical for real-time control tasks such as robotics, where actions must be generated at high frequency.

These limitations motivate the search for more efficient generative policy classes that retain the expressive power of diffusion models while enabling faster inference. Flow matching (Lipman et al., 2023; Liu, 2022) provides one such alternative by replacing stochastic diffusion trajectories with deterministic probability flows. While diffusion models in RL have been the subject of several recent surveys (Zhu et al., 2024; Xu et al., 2025), Flow Matching (FM) in RL represents a distinct mathematical paradigm, rooted in deterministic Optimal Transport (Villani et al., 2009; Peyré & Cuturi, 2019; Lipman et al., 2023) and Ordinary Differential Equations (ODEs), that needs its own dedicated treatment.

The motivation for this survey is threefold:

1. **Algorithmic Distinctiveness:** Unlike diffusion models, which rely on stochastic differential equations (SDEs) and iterative denoising, flow matching operates via deterministic vector field regression. This difference necessitates unique considerations for training stability, inference efficiency, and integration with RL objectives (e.g., Q-learning or Policy Gradients) that are not covered in existing diffusion literature.

2. **Rapid but Fragmented Growth:** As noted, the application of flow matching to control tasks has grown rapidly across disparate subfields. Researchers in offline RL often frame FM as a supervised regression problem, while those in online RL view it as a policy parameterization for actor-critic methods. This survey aims to bridge these communities.

3. **Practical Efficiency:** For real-world robotics and control, the inference speed of generative policies is a critical bottleneck. Flow matching offers a uniquely direct path to acceleration (via ODE solvers and straight flow lines), making it a particularly high-impact area for applied RL researchers.

## 1.2 Positioning Relative to Existing Surveys

While several recent reviews have examined the intersection of generative modeling and reinforcement learning, none provide a focused and unified analysis of the emerging *flow-matching in reinforcement learning paradigm.* This survey addresses this gap by distinguishing flow-based policies from broader diffusion methods and highlighting their unique theoretical and practical advantages.

**Relation to Diffusion RL Surveys.** Foundational surveys such as Zhu et al. (2024) and Xu et al. (2025) establish valuable taxonomies for diffusion-based RL analyzing stochastic diffusion processes (SDEs) and iterative denoising and also categorizing flow matching as an efficient sampling subroutine. We instead provide a detailed look at flow matching in RL from the *deterministic vector-field perspective* that simplifies policy regression and enables direct optimal transport connections.

**Relation to Efficient Generative Modeling.** Surveys on efficient diffusion models (Shen et al., 2025; Ma et al., 2025) focus on accelerating inference via architectural optimizations and fast solvers. We focus on reinforcement learning objectives, offering insights into how flow trajectories can be guided by value functions or shaped by Bellman updates.

**Relation to RL for Generative AI.** Finally, reviews on reinforcement learning for generative AI (Cao et al., 2025; Franceschelli & Musolesi, 2024) emphasize RL as a tool for *aligning* pre-trained models (e.g., RLHF). In contrast, this survey centers on flow matching as a *primary policy learning framework*, where the generative model itself acts as the decision-making agent in sequential control tasks.

**Scope and Organization of this Survey.** This paper focuses specifically on *Continuous Normalizing Flows* and *Flow Matching* approaches applied to sequential decision-making problems, unfolding the literature as a continuous narrative. We begin in Section 2 by establishing the **theoretical foundations**, providing a self-contained introduction to flow matching and its geometric roots in Optimal Transport, explicitly contrasting this deterministic perspective with stochastic diffusion models. Building on this theory, we introduce a **unified taxonomy** in Section 3 to organize the landscape before diving into specific algorithmic paradigms. Section 4 explores the realm of **offline learning**, reviewing how flow matching is utilized for behavioral cloning and offline RL through sophisticated value-guided weighting techniques. The narrative then transitions to **online adaptation** in Section 5, where we examine how agents fine-tune flow policies via active environment interaction, including stochasticized flow policies, pathwise or ODE-based policy gradients, and GRPO-style updates. Section 6 broadens the scope beyond immediate action selection, discussing the use of flows for **critics, environment dynamics, preference alignment, and trajectory generation**. Section 7 then identifies **emerging directions** such as discrete and non-Euclidean action spaces, multi-agent settings, and alignment-oriented extensions. Section 8 summarizes representative **application domains**, while Section 9 reviews common **benchmarks, evaluation protocols, and practical metrics**. Section 10 **compares flow-based methods with Gaussian and diffusion baselines** both conceptually and empirically, summarizing their differences in expressivity, sampling efficiency, and optimization stability while also reporting representative performance numbers across benchmark tables. Finally, Section 11 outlines key **open challenges**, and Section 12 summarizes the **key takeaways** of the survey. Ultimately, the

goal of this survey is to provide both a theoretical primer for newcomers and a practical guide for researchers looking to implement or extend flow-based algorithms in reinforcement learning.

## 2 Background

This section provides the necessary background and notations in reinforcement learning, diffusion models, and flow-matching generative modeling.

### 2.1 Reinforcement Learning Preliminaries

We consider the standard reinforcement learning (RL) framework (Sutton & Barto, 2018). A Markov decision process (MDP) is defined by the tuple $\mathcal{M} = (\mathcal{S}, \mathcal{A}, P, r, \gamma)$, where $\mathcal{S}$ is the state space, $\mathcal{A}$ is the action space, $P(s'|s, a)$ is the transition dynamics, $r(s, a)$ is the reward function, and $\gamma \in (0, 1)$ is the discount factor.

A policy $\pi(a|s)$ induces a distribution over trajectories $\tau = (s_0, a_0, s_1, a_1, \dots)$. The discounted return of a trajectory is

$$G(\tau) = \sum_{t=0}^{\infty} \gamma^t r(s_t, a_t). \tag{1}$$

The associated value and action-value functions are defined as:

$$V^\pi(s) = \mathbb{E}_\pi \left[ \sum_{t=0}^{\infty} \gamma^t r(s_t, a_t) \,\middle|\, s_0 = s \right], \tag{2}$$

$$Q^\pi(s, a) = r(s, a) + \gamma \, \mathbb{E}_{s' \sim P(s'|s,a)}[V^\pi(s')]. \tag{3}$$

**Online Reinforcement Learning.** In the online setting, the agent interacts with the environment to collect trajectories and iteratively improve its policy. The objective is to maximize the expected discounted return:

$$J(\pi) = \mathbb{E}_{\tau \sim \pi}[G(\tau)]. \tag{4}$$

A standard route to optimize this objective is via *policy gradient* methods, which update a parameterized policy $\pi_\phi(a|s)$ in the direction

$$\nabla_\phi J(\pi_\phi) = \mathbb{E}_{(s,a) \sim \pi_\phi} \left[ \nabla_\phi \log \pi_\phi(a|s) \, \hat{A}(s, a) \right], \tag{5}$$

where $\hat{A}(s, a)$ is an estimator of the *advantage*, measuring how much better action $a$ is than the policy's baseline behavior at state $s$.

**Proximal Policy Optimization (PPO).** Among on-policy algorithms, Proximal Policy Optimization (PPO) (Schulman et al., 2017) is particularly important because it will reappear repeatedly in later sections on online flow-based RL. PPO improves stability by replacing the vanilla policy gradient objective with a clipped surrogate objective:

$$\mathcal{L}_{\text{PPO}}(\phi) = \mathbb{E} \left[ \min \left( r_t(\phi)\hat{A}_t, \; \text{clip}(r_t(\phi), 1 - \epsilon, 1 + \epsilon)\hat{A}_t \right) \right], \tag{6}$$

where

$$r_t(\phi) = \frac{\pi_\phi(a_t|s_t)}{\pi_{\phi_{\text{old}}}(a_t|s_t)}$$

is the importance ratio between the current and previous policies, and $\epsilon$ is a small trust-region parameter. Intuitively, PPO prevents overly large policy updates by clipping the incentive to increase or decrease action probability too aggressively. This makes PPO a natural baseline when adapting expressive generative policies, including flow-matching policies, to online RL.

**Group Relative Policy Optimization (GRPO).** A more recent variant, Group Relative Policy Optimization (GRPO) (Shao et al., 2024), replaces the explicit value-function baseline with a *group-relative* advantage estimate. Instead of learning a separate critic, GRPO samples a group of candidate actions or outputs $\{a_i\}_{i=1}^G$ for the same context or state and computes a normalized relative advantage:

$$\hat{A}_i = \frac{R_i - \text{mean}(\{R_j\}_{j=1}^G)}{\text{std}(\{R_j\}_{j=1}^G)}, \tag{7}$$

where $R_i$ is the reward assigned to the $i$-th sample. The policy is then updated using a PPO-style clipped objective, but with these group-normalized advantages instead of critic-based estimates. In other words, GRPO is a *critic-free* relative-ranking variant of policy optimization: it encourages the policy to increase the probability of samples that are better than their group average and decrease the probability of worse ones. This idea has become especially influential in fine-tuning large generative models, and several flow-based online RL methods surveyed later adopt GRPO-style updates.

**Remarks for Flow-Based Policies.** When the policy $\pi_\phi$ is parameterized by a flow model, evaluating $\log \pi_\phi(a|s)$ or the likelihood ratio $r_t(\phi)$ in equation 6 can be substantially more difficult than for Gaussian policies. This is because the policy is defined implicitly through an ODE or continuous normalizing flow, often requiring either differentiation through the ODE solver, likelihood approximations, or alternative surrogate objectives (Chen et al., 2018). Much of the online flow-RL literature can be understood precisely as designing tractable PPO or GRPO-style updates for these expressive but computationally challenging policy classes.

**Offline Reinforcement Learning.** In offline RL (Levine et al., 2020), the agent learns from a fixed dataset $\mathcal{D} = \{(s, a, r, s')\}$ collected by a behavior policy $\mu(a|s)$, without further interaction. Policy improvement is often formulated as a constrained optimization problem, maximizing Q-values while penalizing deviation from the behavior policy $\mu$:

$$\max_\pi \ \mathbb{E}_{s \sim \mathcal{D}} \left[ \mathbb{E}_{a \sim \pi(\cdot|s)} [\, Q(s, a) \,] - \frac{1}{\beta} \, \text{KL}(\pi(\cdot|s) \, \| \, \mu(\cdot|s)) \right]. \tag{8}$$

The closed-form solution is the energy-based policy $\pi^\star(a|s) \propto \mu(a|s) \exp(\beta Q(s, a))$, which forms the basis for many generative RL methods. This solution assumes access to the true $Q$-function and serves as an idealized target for generative policy learning.

## 2.2 Diffusion Models in Reinforcement Learning

Diffusion models (Ho et al., 2020; Song et al., 2021) generate data by learning to reverse a gradual noising process. During training, a forward process progressively corrupts data samples into noise, and a neural network is trained to reverse this process by predicting either the added noise or the score (the gradient of the log-density). At inference time, generation proceeds by iteratively denoising a random noise sample through many small steps, typically formulated as either a discrete reverse chain or a stochastic differential equation (SDE).

Diffusion models (DMs) have become a major paradigm for reinforcement learning because they can represent rich, multi-modal conditional distributions over *actions*, *action sequences*, and even *future trajectories*. These models address the central limitation of unimodal Gaussian policies: the inability to capture diverse, valid behaviors in offline datasets or complex control tasks.

In diffusion-based RL, decision making is implemented by an iterative denoising process that gradually transforms noise into an action (policy view) or an action sequence/trajectory (planning view). Based on recent related survey papers (Zhu et al., 2024; Xu et al., 2025), we provide a categorization of how diffusion models are integrated into RL in Appendix A.

While diffusion policies have achieved strong, and in some cases state of the art performance in offline RL, they remain challenging for control because high-quality generation typically requires tens or hundreds of iterative denoising steps per action, leading to substantial inference latency that can be prohibitive for real-time decision making (Wang et al., 2023; Hansen-Estruch et al., 2023). In addition, the reverse denoising

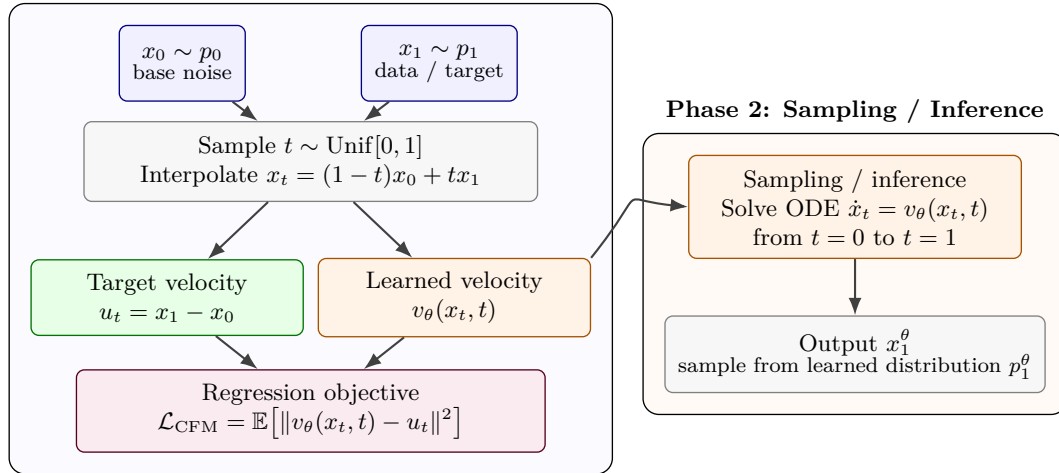

During training, flow matching uses direct regression on sampled interpolation points and *does not require solving the ODE.*

Figure 1: **Flow Matching in a nutshell.** *Top:* flow matching learns approximately straight probability transport from a simple base distribution to the target data distribution. *Bottom left:* training is simulation-free: one samples $(x_0, x_1, t)$, constructs an interpolation point $x_t$, and regresses the learned velocity field $v_\theta(x_t, t)$ toward the target conditional velocity $u_t$. For the common linear optimal-transport path, the target velocity is simply $x_1 - x_0$. *Bottom right:* after training, new samples are generated by integrating the learned ODE from base noise to the target distribution.

process follows a stochastic trajectory in data space, which can make temporal consistency, controllability, and value-based optimization more difficult.

## 2.3 Flow Matching

Flow Matching (FM) (Lipman et al., 2023; Liu, 2022) provides a principled and computationally efficient framework for training Continuous Normalizing Flows (CNFs) (Chen et al., 2018), which model probability transport using ordinary differential equations. Unlike diffusion models, which learn to reverse a stochastic noising process, FM directly learns a deterministic transport map between a simple base distribution and a complex target distribution.

Figure 1 provides a visual summary of this idea. At a high level, flow matching learns a velocity field that transports samples from a simple base distribution to the target distribution through approximately straight probability paths, while keeping training simulation-free and ODE integration is done for inference.

**Simulation-free training.** A key advantage of flow matching is that training is *simulation-free*, meaning that it does not require solving the ODE defined by the model during optimization. Instead, the training objective is formulated as a supervised regression problem on sampled interpolation points $(x_0, x_1, t)$, where the model directly learns the target velocity field. This contrasts with likelihood-based training of continuous normalizing flows or diffusion models, which typically require simulating the forward or reverse process during training.

**Continuous Normalizing Flows.** Let $p_0(x)$ denote a simple base distribution (typically a standard multivariate Gaussian normal distribution $\mathcal{N}(0, I)$ with mean 0 and identity covariance $I$) and $p_1(x)$ the target data distribution. A continuous normalizing flow defines a time-dependent transformation $x_t$ governed by an ordinary differential equation (ODE):

$$\frac{dx_t}{dt} = v_\theta(x_t, t), \qquad x_0 \sim p_0, \tag{9}$$

where $v_\theta(x, t)$ is a neural network parameterizing a velocity field. Solving this ODE from $t = 0$ to $t = 1$ transports samples from $p_0$ to a learned distribution $p_1^\theta$.

Under mild regularity conditions, the induced density $p_t(x)$ evolves according to the continuity equation:

$$\frac{\partial p_t(x)}{\partial t} + \nabla \cdot \big(p_t(x)v_\theta(x, t)\big) = 0, \tag{10}$$

ensuring conservation of probability mass along the flow.

**The Challenge of CNF Training.** Directly training CNFs via maximum likelihood requires solving the ODE and computing log-determinants through the instantaneous change-of-variables formula (Chen et al., 2018), which is computationally expensive and unstable in high dimensions. Flow Matching circumvents this by converting density learning into a supervised regression problem on vector fields.

**Probability Paths and Conditional Flow Matching.** The key idea in flow matching is to define a family of intermediate distributions $\{p_t(x)\}_{t \in [0,1]}$ that smoothly interpolate between $p_0$ and $p_1$. Instead of learning the marginal velocity field directly, we define a *conditional probability path* $p_t(x \mid x_1)$ connecting noise to an individual data point $x_1 \sim p_1$.

Let $\psi_t(x_0, x_1)$ denote a deterministic interpolation between $x_0 \sim p_0$ and $x_1 \sim p_1$. The marginal path is:

$$p_t(x) = \int p_t(x \mid x_1) \, p_1(x_1) \, dx_1.$$

The conditional velocity field associated with this path is:

$$u_t(x \mid x_1) = \frac{d}{dt}\psi_t(x_0, x_1)\Big|_{x=\psi_t(x_0, x_1)}.$$

Flow Matching trains $v_\theta$ to match this conditional velocity in expectation:

$$\mathcal{L}_{\text{CFM}}(\theta) = \mathbb{E}_{\substack{t \sim \text{Unif}[0,1], \\ x_0 \sim p_0, \\ x_1 \sim p_1}} \Big[ \|v_\theta(\psi_t(x_0, x_1), t) - u_t(\psi_t(x_0, x_1) \mid x_1)\|^2 \Big]. \tag{11}$$

Crucially, this objective does *not* require solving the ODE during training. It only requires sampling $(x_0, x_1, t)$ and performing standard regression. Lipman et al. (2023) prove that minimizing this objective yields a velocity field whose induced marginal flow matches $p_1$ exactly under ideal optimization. As illustrated in Figure 1, the training pipeline samples $(x_0, x_1, t)$, constructs an interpolation point $x_t$, and regresses the learned velocity toward the target conditional velocity, rather than solving the full ODE during optimization. Training is *simulation-free* meaning that equation 11 can be evaluated using only samples $(x_0, x_1, t)$ without integrating the ODE, turning density learning into a standard regression problem.

**Optimal Transport and Rectified Flows.** A particularly important choice which is adopted in nearly all RL applications is the linear (optimal transport) interpolation:

$$\psi_t(x_0, x_1) = (1 - t)x_0 + tx_1.$$

In this case, the conditional velocity is constant:

$$u_t(x \mid x_1) = x_1 - x_0.$$

---

**Algorithm 1** Conditional Flow Matching (CFM) / Rectified Flow (OT path) training

---

1: **Input:** dataset distribution $p_1(x)$, base distribution $p_0(x)$ (e.g., $\mathcal{N}(0, I)$), stepsize $\eta$
2: **Initialize:** velocity network $v_\theta(x, t)$
3: **while** not converged **do**
4:     Sample $x_1 \sim p_1(x)$ and $x_0 \sim p_0(x)$
5:     Sample $t \sim \text{Unif}[0, 1]$
6:     **(OT / Rectified path)** set $x_t \leftarrow (1 - t)x_0 + tx_1$
7:     **Target velocity** $u_t \leftarrow x_1 - x_0$                ▷ constant along the linear path
8:     Update $\theta \leftarrow \theta - \eta \nabla_\theta \|v_\theta(x_t, t) - u_t\|_2^2$
9: **end while**
10: **Output:** learned velocity field $v_\theta(x, t)$

---

This choice corresponds to learning straight-line transport between noise and data and is closely related to the Rectified Flow framework (Liu et al., 2023). Rectified flows can be interpreted as learning a marginal-preserving transport that minimizes path curvature, resulting in nearly straight trajectories. This geometric property is one of the main reasons flow models admit accurate low-step ODE solvers during inference.

**Inference.** After training, sampling proceeds by solving the learned ODE:

$$\frac{dx_t}{dt} = v_\theta(x_t, t), \qquad x_0 \sim p_0, \tag{12}$$

using a numerical solver (e.g., Euler or Runge-Kutta). Because the learned paths are approximately straight, accurate sampling is often possible with as few as 1–5 function evaluations which is a major advantage over diffusion models, which often require tens or hundreds of denoising steps.

---

**Algorithm 2** Sampling with a trained flow (ODE integration)

---

1: **Input:** learned velocity $v_\theta(\cdot, \cdot)$, base distribution $p_0$, number of steps $K$
2: Sample $x_0 \sim p_0$
3: Discretize $0 = t_0 < t_1 < \cdots < t_K = 1$ with $\Delta t_k = t_{k+1} - t_k$
4: **for** $k = 0, 1, \ldots, K - 1$ **do**
5:     $x_{t_{k+1}} \leftarrow x_{t_k} + \Delta t_k v_\theta(x_{t_k}, t_k)$           ▷ Euler or RK methods can be used
6: **end for**
7: **Return:** $x_1 \approx x_{t_K}$

---

Conditional Flow Matching forms the mathematical backbone of nearly all flow-based RL methods surveyed in this paper, including energy-weighted offline methods, ODE-differentiated policy gradients, and GRPO-style fine-tuning approaches.

### 2.4 High-level summary of methods for FM on RL

To use flow matching for control, we need to bridge the gap between *likelihood maximization* (learning to generate realistic samples from data) and *reward maximization* (learning to generate actions that achieve high return). At a high level, the existing literature can be organized into two broad families, depending on *where* the reinforcement learning signal enters the flow model. This high-level organization is summarized in Figure 2. The central distinction is whether the RL signal enters the flow model at *training time*, as in offline value or reward weighted regression, or at *policy-update time*, as in online fine-tuning through policy gradients, stochasticized flows, or GRPO-style updates.

**1. Training-Time Guidance (Weighted Regression).** Predominant in **Offline RL**, where one learns from a fixed dataset, this approach embeds the RL objective into the generative training loss. Starting from the KL-regularized offline RL objective in equation 8, the corresponding optimal policy can be written explicitly as

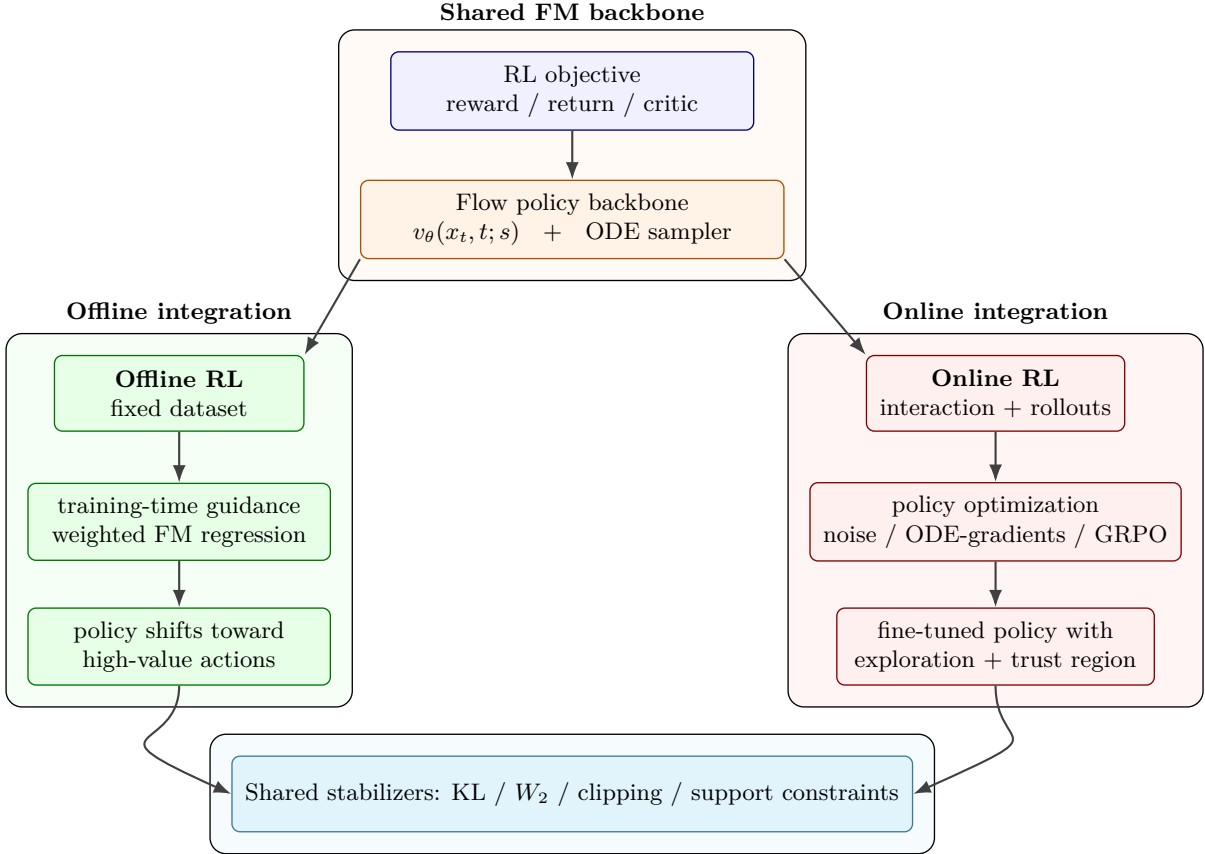

Figure 2: **How flow matching is integrated with reinforcement learning.** A common flow-policy backbone defines an expressive action generator through a learned velocity field and ODE sampler. In *offline RL*, the reinforcement signal typically enters at *training time*, most often by reweighting the flow-matching objective using value or energy estimates. In *online RL*, the signal enters through *policy optimization* using environment interaction, typically via stochasticized flows, pathwise gradients through the ODE, or group-based updates such as GRPO. Across both settings, additional stabilizers such as KL regularization, Wasserstein anchoring, clipping, and support constraints are often used.

$$\pi^\star(a|s) = \frac{\mu(a|s)\exp(\beta Q(s,a))}{\int \mu(a'|s)\exp(\beta Q(s,a'))\,da'},$$

where the denominator is a normalization constant ensuring that $\pi^\star(\cdot|s)$ integrates to one. Hence the closed-form solution is $\pi^\star(a|s) \propto \mu(a|s)\exp(\beta Q(s,a))$ where $\mu(a|s)$ is the behavior policy. Relative to this behavior distribution, actions with larger $Q(s,a)$ values (high-value actions) are exponentially upweighted.

Flow-based offline RL methods use this principle to modify the conditional flow-matching objective so that the learned velocity field preferentially transports noise toward high-value regions of the action space. A generic form is

$$\mathcal{L}_{\text{Weighted-FM}}(\theta) = \mathbb{E}_{(s,a)\sim\mathcal{D},\,t\sim\text{Unif}[0,1]}\left[w(s,a)\left\|v_\theta(x_t,s,t) - u_t(x_t|a,s)\right\|^2\right], \tag{13}$$

where $u_t$ is the target conditional velocity field and the weight function $w(s,a)$ is typically chosen to increase with $Q(s,a)$ or an advantage estimate. Representative examples include Energy-Weighted Flow Matching (EFM / QIPO), which derives a principled Q-guided flow objective (Zhang et al., 2025b), FlowQ, which reformulates energy guidance through analytically tractable path constructions (Alles et al., 2025), and

Flow Q-Learning (FQL), which combines a flow prior with a separate policy-improvement stage (Park et al., 2025b). We discuss this family of approaches in detail in Section 4.

**2. Sampling-Time Guidance (Policy Gradients).** Predominant in **Online RL**, this approach keeps the generative model fixed (or pre-trained) and biases the sampling process. In diffusion, this is often done via classifier guidance $\nabla_x Q(s, x)$. In flow matching, one can differentiate the Q-value with respect to the generated action and backpropagate gradients through the ODE solver to update the flow parameters $\theta$ directly.

In online RL, where the flow model acts directly as the policy and is updated through interaction with the environment, the key question is how to optimize a flow-defined policy under policy-gradient style objectives despite the fact that actions are generated implicitly by integrating an ODE.

At a conceptual level, if the generated action is written as

$$a = \Phi_\theta(z, s), \qquad z \sim p_0,$$

where $\Phi_\theta$ denotes the solution map of the flow ODE

$$\frac{dx_t}{dt} = v_\theta(x_t, s, t), \qquad x_0 = z,$$

and the generated action corresponds to the terminal state $a = x_1$. Therefore $\Phi_\theta$ denotes the ODE solution map induced by the learned velocity field, then the policy objective can be differentiated through the sampling procedure:

$$\nabla_\theta J(\pi_\theta) = \mathbb{E}_{z \sim p_0} \left[ \nabla_a Q(s, a) \frac{\partial \Phi_\theta(z, s)}{\partial \theta} \right]. \tag{14}$$

In practice, existing methods make this idea tractable in different ways: some inject noise into the flow dynamics so that standard likelihood-based methods such as PPO can be used (Zhang et al., 2025c); some backpropagate policy gradients directly through the ODE solver or use flow-matching losses as PPO-compatible surrogates (McAllister et al., 2025; Pfrommer et al., 2025; Yang et al., 2026); and others adopt group-based online optimization strategies such as GRPO-style updates (Liu et al., 2025a; He et al., 2025; Yu et al., 2026; Chen et al., 2025c; Sun et al., 2025). We discuss this family of approaches in detail in Section 5.

**Summary.** Flow matching occupies a unique position among generative policy classes: it combines the expressiveness of diffusion models with deterministic sampling paths, enabling both efficient inference and stable optimization. In offline RL, this allows reward-weighted regression objectives that avoid backpropagation through long sampling chains. In online RL, deterministic ODEs permit direct differentiation through the policy with reduced variance compared to stochastic diffusion trajectories. These properties make flow matching a natural bridge between classical RL objectives and modern generative modeling. Figure 2 also highlights that, despite their differences, both offline and online flow-based RL methods share the same flow-policy backbone and often rely on related stabilizers such as KL regularization, Wasserstein anchoring, clipping, or support constraints.

## 3   A Taxonomy of Flow Matching in Reinforcement Learning

Flow matching has rapidly emerged as a principled alternative to diffusion models for decision-making. Unlike diffusion, which relies on stochastic denoising, flow matching learns deterministic transport maps defined by velocity fields (Lipman et al., 2023; Liu, 2022). In RL, this enables efficient sampling, stable differentiation, and direct integration with value-based objectives.

To organize the rapidly growing literature, we propose a taxonomy based on two orthogonal design axes:

1. **What distribution is modeled?** (Actions, Critics, Dynamics, or Preferences).

2. **Where does the RL signal enter?** (Training-time weighted regression, ODE/pathwise gradients, or Group-relative optimization).

Figure 3 visualizes this landscape, categorizing the methods discussed in this survey.

**Taxonomy of Flow-Based RL Methods.** Throughout this survey, we organize the literature as follows. Sections 4 and 5 categorize methods according to the *learning setting* in which the flow policy is optimized (offline versus online reinforcement learning). Section 6, in contrast, broadens the perspective to consider *what object is modeled by the flow*, including critics, environment dynamics, and preference alignment objectives. This organization allows us to highlight both the algorithmic differences between offline and online learning as well as the expanding role of flow matching beyond policy parameterization.

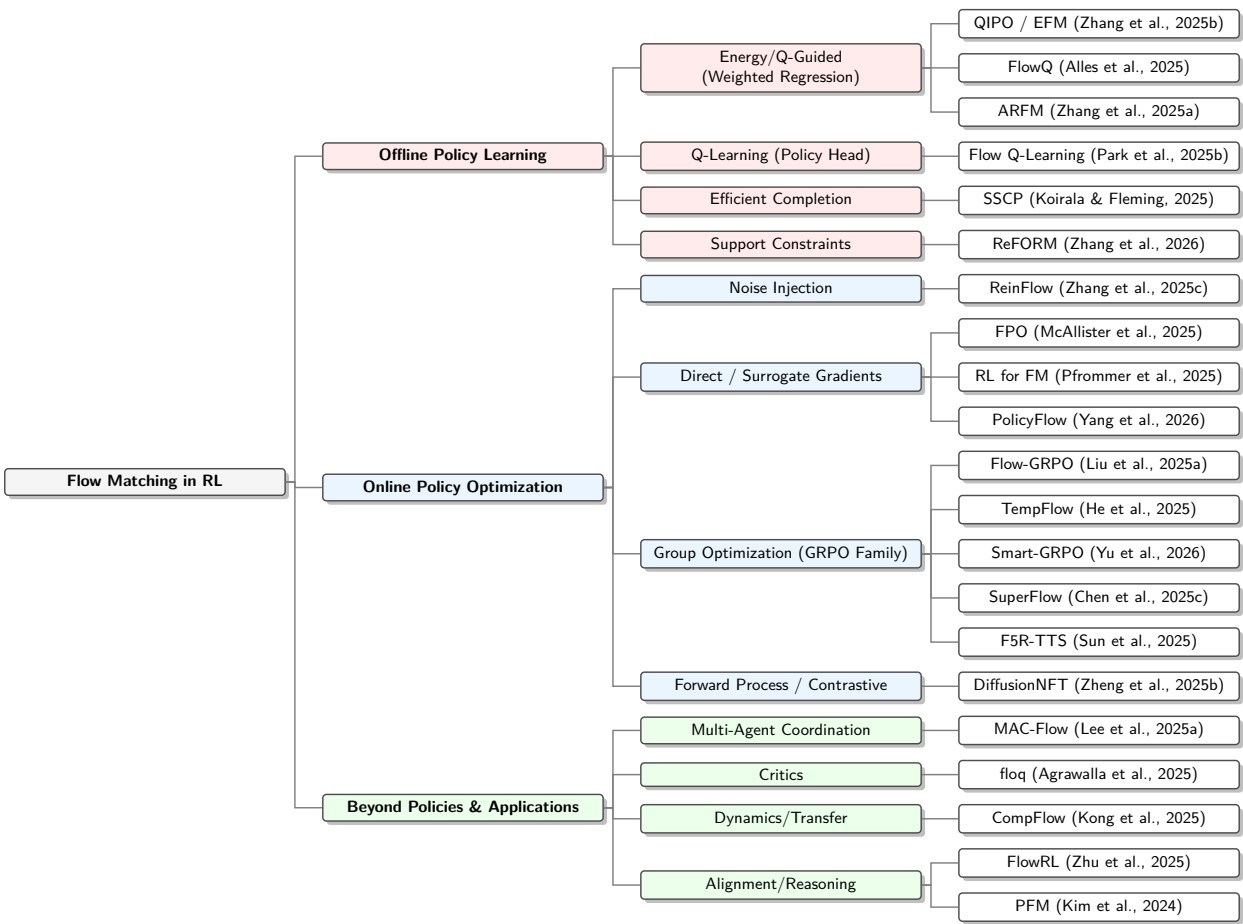

Figure 3: **A Taxonomy of Flow Matching in Reinforcement Learning.** We categorize methods by their primary function: Offline Policy Learning (Section 4), Online Policy Optimization (Section 5), and Applications Beyond Policy Learning (Section 6).

**Two-axis cross summary.** While Sections 4–6 are organized along the learning-setting axis (offline / online / beyond) for narrative clarity, Table 1 explicitly crosses the two axes introduced above: *what distribution is modeled* (rows) versus *where the RL signal enters* (columns). The resulting 2D view makes gaps in the literature visible (e.g., GRPO-style updates have been almost exclusively applied to action and preference flows, weighted-regression critics and dynamics flows are still rare), and it highlights that several methods occupy hybrid cells (e.g., FQL combines a behavior-cloning flow with a separate one-step actor head).

## 3.1 Axis A: What Distribution is Modeled?

**Action (Policy) Flows.** The most common application is modeling conditional action distributions $\pi(a|s)$. The flow maps a base distribution (e.g., Gaussian noise) to multimodal actions. This paradigm underlies both offline methods like *FlowQ* (Alles et al., 2025) and *Flow Q-Learning* (Park et al., 2025b), as well as

| What is modeled (Axis A) | How the RL signal enters (Axis B) | | | |
|---|---|---|---|---|
| | Weighted regression | Direct / surrogate gradients | Group-relative (GRPO) | Other mechanisms |
| **Actions / policies** | QIPO/EFM (Zhang et al., 2025b), FlowQ (Alles et al., 2025), ARFM (Zhang et al., 2025a), ORW-CFM-W2 (Fan et al., 2025b), ADRPO (Fan et al., 2025c), RWFM (Pfrommer et al., 2025) | FPO (McAllister et al., 2025), PolicyFlow (Yang et al., 2026), RL-FM (Pfrommer et al., 2025) | Flow-GRPO (Liu et al., 2025a), TempFlow (He et al., 2025), Smart-GRPO (Yu et al., 2026), SuperFlow (Chen et al., 2025c), F5R-TTS (Sun et al., 2025) | FQL (Park et al., 2025b), SSCP (Koirala & Fleming, 2025), ReFORM (Zhang et al., 2026), MAC-Flow (Lee et al., 2025a) *(distillation)*, ReinFlow (Zhang et al., 2025c) *(noise injection)*, DiffusionNFT (Zheng et al., 2025b) *(forward-process)* |
| **Critics / values** | floq (Agrawalla et al., 2025) *(TD regression on flow critic)* | — | — | DFC (Chen et al., 2025a), FlowCritic (Zhong et al., 2025), EVOR (Espinosa-Dice et al., 2025) *(distributional flow critics)* |
| **Dynamics / transitions** | CompFlow (Kong et al., 2025) *(transport regression on transitions)* | — | — | — |
| **Preferences / alignment** | PFM (Kim et al., 2024), FlowRL (Zhu et al., 2025) | — | Fine-Grained GRPO (Zhou et al., 2025), AWM (Xue et al., 2025), GRPO-Guard (Wang et al., 2025b) | AC-Flow (Fan et al., 2025a) *(intermediate feedback)* |

Table 1: **Two-axis cross summary of flow-matching methods in RL.** Rows indicate what distribution the flow models (Axis A) and columns indicate how the RL signal enters the flow training (Axis B). Empty cells mark under-explored combinations that represent open research directions such as GRPO-style updates for critic or dynamics flows. This table complements the learning-setting view in Figure 3 and the offline/online/beyond narrative in Sections 4–6.

online methods like *Flow-Based Policy* (Lv et al., 2025). Recent works such as *SSCP* (Koirala & Fleming, 2025) focus on making these action flows efficient via single-step completion. Extending beyond single agents, flow matching can model the complex joint action distributions of multiple interacting agents. *MAC-Flow* (Lee et al., 2025a) learns a centralized flow-based representation of joint behaviors from offline data, which is then distilled into decentralized one-step policies for fast, independent execution.

**Critic Flows.** A radical departure from standard RL is modeling the *Q-function itself* as a flow. *floq* (Agrawalla et al., 2025) trains a flow to represent the distribution of returns, allowing for "test-time compute" scaling in value estimation; running the critic flow longer yields more precise value estimates.

**Dynamics and Transition Flows.** Instead of policies, flow matching can model environment transitions $P(s'|s, a)$. *Composite Flow Matching (CompFlow)* (Kong et al., 2025) uses this to estimate the "dynamics gap" between source and target environments, enabling principled transfer learning under shifted dynamics.

**Preference and Reasoning Flows.** In alignment tasks, flows model the distribution of "preferred" outputs (e.g., reasoning chains or speech). *FlowRL* (Zhu et al., 2025) and *F5R-TTS* (Sun et al., 2025) use flows to balance diversity and reward maximization in LLMs and Text-to-Speech, respectively.

## 3.2 Axis B: Where Does the RL Signal Enter?

The second axis distinguishes methods by *how* the reinforcement learning objective enters the flow model. As introduced in Section 2.4, this axis admits three broad categories: (i) *training-time weighted regression*, which embeds the RL signal into the generative loss by reweighting the flow-matching objective with energy or advantage terms (Section 4); (ii) *online policy gradients*, which introduce the RL signal through environment interaction via noise injection, ODE backpropagation, or surrogate objectives (Section 5); and (iii) *group-relative optimization*, which ranks sampled candidates by reward and updates the flow accordingly (also Section 5).

# 4 Offline RL with Q/Energy-Guided Flow Policies

Offline reinforcement learning learns a policy from a fixed dataset $\mathcal{D} = \{(s, a, r, s')\}$ without further interaction, which makes *distributional shift* the central challenge: if a learned policy proposes actions outside the dataset support (out-of-distribution (OOD) actions), value learning (the learned Q-function) may extrapolate inaccurately, leading to severe overestimation errors. Recent work successfully integrates flow matching into offline RL to simultaneously model highly expressive, multi-modal behavior distributions while safely incorporating value-based policy improvement. The unifying principle across these methods is to learn a generative flow policy whose target distribution is an *energy-reweighted* version of the behavior distribution. These methods can be viewed as special cases of the general guided-flow framework introduced by Feng et al. (2025a), which interprets value guidance as a modification of the flow velocity field which we detail in Section 6.4.

## 4.1 Regularized Policy Optimization as an Energy-Weighted Target

To prevent OOD actions, many offline RL algorithms constrain the learned policy $\pi$ to remain close to the behavior policy $\pi_\beta$ using a Kullback-Leibler (KL) divergence penalty. KL-regularized offline RL defines an ideal policy improvement target as an energy-reweighted behavior distribution. A common way to formalize "stay close to data while improving" is the following regularized optimization problem:

$$\max_\pi \ \mathbb{E}_{s \sim \mathcal{D}} \left[ \mathbb{E}_{a \sim \pi}[Q(s, a)] - \frac{1}{\beta} D_{\mathrm{KL}}\big(\pi(\cdot|s) \,\|\, \pi_\beta(\cdot|s)\big) \right],$$

where $\beta > 0$ acts as an inverse temperature parameter controlling the strength of the behavioral penalty. The closed-form optimal solution to this objective (when $Q$ and $\pi_\beta$ are known) is an *energy-weighted distribution*:

$$\pi^*(a|s) = \frac{\pi_\beta(a|s) \exp(\beta Q(s, a))}{Z(s)}, \tag{15}$$

where $Z(s)$ is the intractable partition function (normalizing constant). While this formulation is theoretically nice, direct sampling from $\pi^*(a|s)$ is not possible because the true density $\pi_\beta$ is unknown (only accessible through samples, not density evaluation), and the partition function cannot be efficiently computed in continuous action spaces.

Flow matching provides a simulation-free generative framework to directly approximate and sample from this optimal energy-weighted distribution by making use of equation 15 through learning a transport map that can sample from the reweighted target without evaluating partition functions; equation 15 which can be read as an *energy-based* distribution with energy $E(s, a) = -Q(s, a)$ provides a clean "north star" for generative policy learning, because the role of the generative model becomes *sampling from an energy-reweighted dataset distribution* rather than directly optimizing a conventional actor network.

Flow-based offline RL therefore proceeds in two conceptual steps: first learn a high-capacity conditional flow that models the behavior support (a strong $\pi_\beta$ surrogate), and then incorporate the energy term $\exp(\beta Q)$ to tilt the learned generative process toward higher-value actions. The remainder of this section can be read as a sequence of increasingly careful answers to the same question: *how should Q enter the flow training/inference so that improvement happens, but OOD actions do not?*

## 4.2 Energy-Weighted Flow Matching (EFM / QIPO)

Energy-Weighted Flow Matching (EFM) (Zhang et al., 2025b), also referred to as Q-guided Iterative Policy Optimization (QIPO), formalizes the problem of learning a flow from a source noise distribution $p_0(a|s)$ to the optimal energy-weighted target $p_1(a|s) \propto \pi_\beta(a|s) \exp(\beta Q_\psi(s, a))$. Here, the negative Q-value acts as the energy function: $E(a) = -Q_\psi(s, a)$.

EFM/QIPO learns an energy-guided flow *during training* by turning policy improvement into a weighted vector-field regression problem studying the general problem of learning a guided distribution of the form

$$q(x) \ \propto \ p(x) \exp\big(-\beta E(x)\big), \tag{16}$$

and shows how to learn a flow that produces samples from $q$ *without* learning intermediate guidance models along the path. In offline RL, one instantiates $x = a$ (optionally conditioned on $s$) and chooses the energy as $E(s, a) = -Q_\psi(s, a)$, so the guidance factor becomes $\exp(\beta Q_\psi(s, a))$. The resulting algorithm in Zhang et al. (2025b) is referred to as *Q-weighted Iterative Policy Optimization (QIPO)*, and its distinguishing feature is that policy improvement is performed *without backpropagating through an ODE solver or sampling chain*: the policy update remains a supervised regression update.

Here the key technical obstacle is that the guided marginal path depends on an intractable *intermediate energy* $E_t(x)$. To learn a flow, conceptually choose a base probability path $\{p_t\}_{t \in [0,1]}$ (e.g., the OT/linear path used in conditional flow matching) and seek a *guided* path

$$q_t(x) \; \propto \; p_t(x) \exp\big( - E_t(x) \big),$$

whose terminal distribution matches equation 16. The difficulty is that the intermediate energy $E_t(x)$ is generally not available in closed form, and naively learning it would reintroduce the auxiliary-model burden that this work aims to avoid.

The authors resolve this by proving that optimizing a simple *energy-weighted* flow-matching loss recovers the guided flow without explicitly modeling $E_t$. Rather than learning $E_t$, they show that one can directly optimize an energy-weighted regression objective that matches the guided flow field in expectation. A convenient view (especially for offline RL) is through the *conditional* form of the objective: samples are drawn from the *unguided* conditional path, while the loss is reweighted by an energy term evaluated at a terminal sample.

**Why this works.** The argument relies on a simple importance-reweighting identity. Suppose we want to match the velocity field of the *guided* marginal path $q_t(x) \propto p_t(x) \exp(-E_t(x))$, but we only have access to samples $(x_0, x_1, t)$ from the *unguided* base path with terminal $x_1 \sim p_1$. Because $q_1(x_1) \propto p_1(x_1) \exp(-E(x_1))$ by construction, samples from $q_1$ can be obtained, in expectation, by reweighting samples from $p_1$ with weights $w(x_1) \propto \exp(-E(x_1))$. Zhang et al. (2025b) extend this idea to the intermediate times where they show that minimizing the conditional flow-matching loss with terminal weights $w(x_1)$ matches the velocity field of the guided marginal $q_t$ in expectation, *without* ever needing to evaluate the intermediate energy $E_t(x)$. Specializing $E(s, a) = -Q_\psi(s, a)$ recovers equation 17 and the softmax over the support set in QIPO implements this reweighting in a partition-function-free, batch-normalized way. We refer interested readers to Zhang et al. (2025b) (their Theorem 4.1 for the closed-form energy-guided velocity field, and Theorem 4.3 for the equivalence of the energy-weighted and conditional energy-weighted losses) for a formal proof.

Specializing the energy to $E(s, a) = -Q_\psi(s, a)$ yields QIPO's policy improvement update:

$$\mathcal{L}_{\text{QIPO}}(\theta) = \mathbb{E}_{\substack{s \sim \mathcal{D}, \, a_1 \sim \pi_\beta(\cdot|s) \\ t \sim \text{Unif}[0,1], \, a_t \sim p_t(\cdot|a_1, s)}} \left[ \bar{w}_\psi(s, a_1) \left\| v_\theta(a_t, s, t) - u_t(a_t | a_1, s) \right\|_2^2 \right], \tag{17}$$

where $u_t$ is the *base* conditional target field (e.g., OT: $u_t = x_1 - x_0$), and $\bar{w}_\psi$ is a *normalized* energy weight. In practice, $\bar{w}_\psi$ is computed by a softmax over a *finite support set* of candidate actions, which avoids computing partition functions and stabilizes the exponential weighting.

The algorithmic backbone of QIPO is an alternating loop of (i) critic learning and (ii) energy-weighted flow regression with support-set normalization. QIPO follows the standard offline RL rhythm: it trains a critic $Q_\psi$ on the dataset using Bellman regression, and periodically performs policy improvement updates that tilt the flow toward high-$Q$ actions. The important implementation detail is that the weights in equation 17 are computed using a *support action set* per state: given a batch of states, one samples $M$ candidate actions (from the current policy or a behavior model), evaluates $Q_\psi(s, a)$ on those candidates, and computes

$$\bar{w}_\psi(s, a^{(j)}) \; = \; \frac{\exp\big(\beta Q_\psi(s, a^{(j)})\big)}{\sum_{k=1}^{M} \exp\big(\beta Q_\psi(s, a^{(k)})\big)}.$$

This "softmax over a support set" is exactly the step that makes QIPO practical in continuous control: it replaces the intractable partition function by a batch-normalized estimator while keeping the update

---

**Algorithm 3** Flow-based QIPO / EFM for Offline RL (Zhang et al., 2025b)

---

**Require:** Offline dataset $\mathcal{D}$; flow policy velocity $v_\theta(a, s, t)$; critic $Q_\psi$ (and target $\bar{\psi}$); support size $M$; temperature $\beta$; CFM path sampler $a_t \sim p_t(\cdot|a_1, s)$; base target field $u_t(\cdot|a_1, s)$ (e.g., OT).
 1: **(Warm-up)** Train $v_\theta$ with standard (unweighted) CFM on $\mathcal{D}$ to model behavior actions.
 2: **(Warm-up)** Train $Q_\psi$ on $\mathcal{D}$ with Bellman regression (e.g., TD target using a policy prior).
 3: **while** not converged **do**
 4:     Sample minibatch $\{(s_i, a_i, r_i, s_i')\}_{i=1}^B \sim \mathcal{D}$.
 5:     **Critic update:** update $\psi$ to minimize a TD/Bellman loss on $\mathcal{D}$ (with target network $\bar{\psi}$).
 6:     **Support set:** for each $s_i$, sample candidate actions $\{a_i^{(j)}\}_{j=1}^M$ from the current flow policy (or a behavior flow prior).
 7:     Compute normalized weights $\bar{w}_i^{(j)} \leftarrow \frac{\exp(\beta Q_\psi(s_i, a_i^{(j)}))}{\sum_{k=1}^M \exp(\beta Q_\psi(s_i, a_i^{(k)}))}$.
 8:     **Flow update:** for each $(s_i, a_i^{(j)})$, sample $t \sim \text{Unif}[0, 1]$ and $a_t \sim p_t(\cdot|a_i^{(j)}, s_i)$.
 9:     Update $\theta$ by minimizing

$$\sum_{i=1}^B \sum_{j=1}^M \lambda(t)\, \bar{w}_i^{(j)} \big\| v_\theta(a_t, s_i, t) - u_t(a_t|a_i^{(j)}, s_i) \big\|_2^2.$$

10: **end while**
11: **Output:** flow policy obtained by integrating $\dot{a}_t = v_\theta(a_t, s, t)$ from $t = 0 \rightarrow 1$ at test time.

---

simulation-free. Algorithm 3 summarizes the backbone of flow-based QIPO/EFM in a form that matches the structure used in Zhang et al. (2025b).

Conceptually, QIPO's "energy guidance" happens by changing *which terminal samples the regression cares about*, not by changing how the flow is numerically integrated. So the ODE solver is not part of training; the only place where $Q$ enters is the weight $\bar{w}$, which selectively emphasizes high-value actions inside the dataset's support. This is precisely why QIPO tends to be stable compared to methods that differentiate through an iterative generative sampler: the policy update is a weighted regression update, and the heavy lifting is done by the critic.

### 4.3 FlowQ: Approximating Energy-Guided Paths

FlowQ (Alles et al., 2025) refines energy-guided training by approximating the *guided probability path* and deriving an explicit guided conditional velocity field. While QIPO/EFM provides a general energy-weighted objective, FlowQ takes a complementary route: it assumes a guided marginal path of the form

$$\hat{p}_t(x) \;\propto\; p_t(x) \exp\{-\lambda(t) E(x)\},$$

and then derives a tractable *conditional* path and velocity field by approximating $E(x_t)$ with a first-order Taylor expansion around the base-path mean. In offline RL, again $E(s, a) = -Q_\psi(s, a)$, so the guidance favors higher Q-values.

The specific step FlowQ refines relative to QIPO is the modeling of *how guidance distorts the intermediate-time conditionals*. The algorithm discussed in the previous section can be interpreted as "keep the base conditional path but weight terminal samples by $\exp(\beta Q)$", which is powerful but leaves the intermediate path implicit. FlowQ instead explicitly constructs an approximate conditional

$$\hat{p}_t(x_t|x_1) \;\propto\; p_t(x_t|x_1) \exp\{-\lambda(t) E(x_t)\},$$

and shows that, under the standard Gaussian CFM conditional $p_t(x_t|x_1) = \mathcal{N}(x_t; tx_1, (1-t)^2 I)$ and a first-order expansion of $E$, the guided conditional remains Gaussian with a *shifted mean*. Concretely, the approximation yields

$$\hat{p}_t(x_t|x_1) \approx \mathcal{N}\big(x_t;\; \alpha_c(t, x_1),\; (1-t)^2 I\big), \qquad \alpha_c(t, x_1) = tx_1 - (1-t)^2 \lambda(t) \nabla E(tx_1),$$

---

**Algorithm 4** Flow Q-Learning (FQL) for Offline RL (Park et al., 2025b)

---

**Require:** Offline dataset $\mathcal{D}$; BC flow velocity $v_\theta(a, s, t)$ and induced flow policy $\mu_\theta(s, z)$ (Euler steps $M$); critic $Q_\phi$ (target $\bar{\phi}$); one-step actor $\mu_\omega(s, z)$; coefficient $\alpha$.

1: **while** not converged **do**
2:      Sample minibatch $\{(s, a, r, s')\} \sim \mathcal{D}$.
3:      **Critic update:** sample $z \sim \mathcal{N}(0, I)$, set $a' \leftarrow \mu_\omega(s', z)$, update $\phi$ using TD loss

$$\big(Q_\phi(s, a) - (r + \gamma Q_{\bar{\phi}}(s', a'))\big)^2.$$

4:      **BC flow update:** sample $x_0 \sim \mathcal{N}(0, I)$, set $x_1 \leftarrow a$, sample $t \sim \mathrm{Unif}[0, 1]$, set $x_t \leftarrow (1 - t)x_0 + tx_1$, update $\theta$ to minimize $\|v_\theta(x_t, s, t) - (x_1 - x_0)\|_2^2$ (OT-CFM).
5:      **One-step actor update:** sample $z \sim \mathcal{N}(0, I)$, set $a_\pi \leftarrow \mu_\omega(s, z)$, update $\omega$ to minimize

$$-Q_\phi(s, a_\pi) + \alpha\|a_\pi - \mu_\theta(s, z)\|_2^2.$$

6: **end while**
7: **Output:** one-step policy $\mu_\omega$ for fast inference; $\mu_\theta$ as an expressive prior.

---

which induces an explicit guided target velocity $\hat{u}_t(\cdot|x_1)$ for regression. This derivation clarifies *what changes*: the authors in this paper propose modifying the *conditional geometry of the path* through a mean shift that pushes mass toward low energy (high $Q$) regions, rather than only reweighting endpoints.

In practice, FlowQ preserves the same "simulation-free" training philosophy while using the guided conditional field in place of the base one. Because the guided conditional is still a simple Gaussian under the approximation, this method can sample $x_t$ cheaply and regress $v_\theta$ onto the derived $\hat{u}_t$ without integrating an ODE during training. The net result is a policy that internalizes guidance at training time, so that inference can remain plain ODE sampling without additional test-time guidance.

### 4.4 Flow Q-Learning: One-Step Improvement on Top of a Flow Prior

Flow Q-Learning (FQL) (Park et al., 2025b) refines offline flow policies by moving the RL objective from the iterative flow sampler to a separate one-step actor. A direct way to apply actor-critic learning to a flow policy is to treat the generated action as $a = \mathrm{ODE}(z; v_\theta, s)$ and backpropagate $\nabla_a Q(s, a)$ through the ODE solver. The authors in this paper argue that this *recursive backpropagation through time* is often unstable and expensive for iterative generative policies, and proposes a modular alternative: train the flow *only* as a behavior model, and train a *one-step* policy to maximize value while being regularized toward the flow prior.

The backbone of their algorithm is a three-part loop: behavior flow training, critic training, and one-step policy improvement with distillation. FQL begins by training a behavior-cloning flow policy $\mu_\theta(s, z)$ with standard CFM, producing an expressive implicit distribution over actions. It then trains a critic $Q_\phi$ on the dataset, and finally trains a one-step actor $\mu_\omega(s, z)$ with a reparameterized policy gradient objective augmented with a *distillation* term that keeps $\mu_\omega$ close to the flow prior:

$$\min_\omega \; \mathbb{E}_{s \sim \mathcal{D}, \, z \sim \mathcal{N}(0, I)} \Big[ -Q_\phi\big(s, \mu_\omega(s, z)\big) + \alpha\big\|\mu_\omega(s, z) - \mu_\theta(s, z)\big\|_2^2 \Big].$$

This design makes the refinement relative to QIPO/FlowQ explicit: instead of changing the flow regression target with $Q$, FQL *keeps the flow regression purely behavioral* and performs value maximization in a separate policy head. In Algorithm 4 we present the core structure of FQL as given in Park et al. (2025b).

In short, FQL replaces "guided generation" with "guided distillation," yielding efficient inference while retaining a rich generative prior. Because the deployed actor is one-step, this method avoids ODE-time inference costs entirely, yet the actor remains expressive because it is trained under a strong flow prior that captures multimodality in the dataset.

### 4.5 ReFORM: On-Support Improvement via Reflected Noise Manipulation

ReFORM (Zhang et al., 2026) refines offline flow RL by enforcing support constraints *structurally* through bounded noise and reflected flows. Many offline RL methods rely on statistical penalties (e.g., KL or Wasserstein) to keep policies close to behavior, but such penalties do not directly guarantee $\text{supp}(\pi) \subseteq \text{supp}(\pi_\beta)$. ReFORM targets the support constraint head-on by designing the policy as a composition of (i) a behavior-cloning flow with *bounded-support* latent noise and (ii) a *noise generator* that manipulates noise *within the same bounded support.* The key guarantee is nice and simple: if the manipulated noise distribution stays inside the original noise support, then pushing it through the BC flow cannot leave the BC policy's action support.

The first pillar of ReFORM is to choose a bounded latent source so that behavior support is approximated by the BC flow's pushforward of that bounded set. Let the BC flow policy be the time-1 map $\psi_{\theta_1}(1, z; s)$ of a flow that transports $z \sim q_{\text{BC}}$ to actions, where $q_{\text{BC}}$ has bounded support (e.g., uniform on a ball). ReFORM uses the observation that $\text{supp}(\pi_\beta(\cdot|s))$ can be approximated by $\psi_{\theta_1}(1, \text{supp}(q_{\text{BC}}); s)$, so staying inside $\text{supp}(q_{\text{BC}})$ in latent space is the lever that controls on-support behavior in action space.

The second pillar of their proposed method is a *reflected* flow for noise manipulation that guarantees the manipulated noise stays within the bounded support. The continuous-time reflected ODE and the discrete projected Euler step used in practice are not mathematically equivalent, and it is important to distinguish them.

**Theoretical construction.** ReFORM models the noise generator as a reflected ODE whose continuous-time dynamics are given by

$$d\psi_{\theta_2}(t, w; s) = v_{\theta_2}\big(t, \psi_{\theta_2}(t, w; s); s\big)\, dt + dL_t,$$

where $L_t$ is a non-decreasing local-time process supported only on the boundary of $\text{supp}(q_{\text{BC}})$. The local-time term acts as an instantaneous inward push whenever the trajectory contacts the boundary, and its mathematical role is to formally guarantee that $\psi_{\theta_2}(t, w; s)$ remains inside $\text{supp}(q_{\text{BC}})$ for all $t$.

**Practical algorithm.** The continuous-time reflection is not simulated directly and instead, ReFORM uses a discretization-time *projected / reflected Euler step* where after each Euler update $\tilde{z}_{k+1} = z_k + h\, v_{\theta_2}(t_k, z_k; s)$, the iterate is projected back onto $\text{supp}(q_{\text{BC}})$ if it crossed the boundary. This discrete projection is a numerical surrogate for the continuous local-time term as it does not exactly reproduce the dynamics of the reflected SDE, but it preserves the key invariant required for support containment, namely that the final iterate lies inside the bounded latent support. The per-step cost is essentially the same as standard Euler. We refer interested readers to Zhang et al. (2026) (their Eq. 9 for the continuous reflected ODE and Eq. 12 for the projected Euler discretization, with Theorem 1 proving the support containment guarantee for the discrete procedure).

ReFORM's optimization is then cleanly separated into behavior modeling, noise optimization for value, and (optionally) one-step distillation for deployment. This method first trains the BC flow $\psi_{\theta_1}$ on the dataset, then trains the constrained noise generator $\psi_{\theta_2}$ to increase $Q$ under the composed policy $\psi_{\theta_1}(1, \cdot; s) \circ \psi_{\theta_2}(1, \cdot; s)$, and finally distills the BC flow into a one-step policy for computational efficiency. The distillation loss follows the same spirit as other flow-distillation approaches:

$$\mathcal{L}_{\text{Distill}}(\hat{\theta}_1) = \mathbb{E}_{s \sim \mathcal{D},\, z \sim q_{\text{BC}}} \big\| \hat{\mu}_{\hat{\theta}_1}(s, z) - \psi_{\theta_1}(1, z; s) \big\|_2^2. \tag{18}$$

Algorithm 5 summarizes the high-level pipeline of this method.

The main conceptual payoff of their proposed algorithm is that it turns "stay on support" from a penalty into a construction. Because support containment is ensured by keeping latent noise inside the bounded source support and using reflected dynamics to prevent boundary escape, ReFORM can pursue policy improvement without relying exclusively on statistical regularizers that may either be too weak to prevent OOD or too strong to prevent improvement.

---

**Algorithm 5** ReFORM for Offline RL (Zhang et al., 2026)

---

**Require:** Offline dataset $\mathcal{D}$; bounded-support latent $q_{\mathrm{BC}}$; BC flow $\psi_{\theta_1}$; reflected noise flow $\psi_{\theta_2}$; critic $Q_\psi$.

1: **Train BC flow:** fit $\psi_{\theta_1}$ (velocity field) by CFM/OT-CFM on $(s, a) \sim \mathcal{D}$ with $z \sim q_{\mathrm{BC}}$.
2: **Train critic:** fit $Q_\psi$ on $\mathcal{D}$ with standard Bellman regression.
3: **while** not converged **do**
4:     Sample states $s \sim \mathcal{D}$ and base noise $w \sim q_{\mathrm{BC}}$.
5:     **Reflected noise step:** generate manipulated noise $\tilde{z} = \psi_{\theta_2}(1, w; s)$ by integrating the reflected ODE (projected/reflected Euler).
6:     **Compose policy:** generate action $a = \psi_{\theta_1}(1, \tilde{z}; s)$.
7:     Update $\theta_2$ to increase $Q_\psi(s, a)$ subject to the reflection constraint (implemented by the reflected dynamics).
8: **end while**
9: **(Optional) Distill:** train one-step policy $\hat{\mu}_{\hat{\theta}_1}$ to match $\psi_{\theta_1}(1, z; s)$ using equation 18.
10: **Output:** on-support policy via composed flows (and optionally one-step distilled policy).

---

## 4.6 Efficiency and Stability Add-ons

Recent work complements the core offline algorithms by addressing inference cost and weight variance without changing the overall story. Even when training is simulation-free, deploying a flow policy can incur ODE integration cost, and exponential Q-weights can have high variance; both issues have motivated practical add-ons that preserve the backbone described above.

Single-step completion distills iterative flow generation into one-shot inference by learning to "complete" the transport from intermediate points. Single-Step Completion Policies (SSCP) (Koirala & Fleming, 2025) modify the supervision signal so that, from an intermediate $a_t$, the model predicts the completion toward $a_1$, enabling inference in a single network evaluation while retaining the representational benefits of flow training.

Adaptive balancing targets the bias-variance tradeoff induced by aggressive exponential weighting in offline post-training. Adaptive Reinforced Flow Matching (ARFM) (Zhang et al., 2025a) addresses the practical instability that arises when $\exp(\beta Q)$ weights become overly peaked by adapting the effective scaling of the weight to preserve usable learning signal while controlling gradient variance, which is particularly relevant in large-scale post-training settings.

**Summary.** Across QIPO/EFM, FlowQ, FQL, and ReFORM, the offline literature can be read as a sequence of increasingly careful design choices about *where* guidance should enter: QIPO/EFM tilts the regression objective with normalized Q-weights, FlowQ explicitly models how guidance distorts intermediate-time conditionals, FQL relocates value maximization to a one-step actor regularized by a flow prior, and ReFORM enforces on-support improvement structurally through bounded latent noise and reflected flows. Table 2 synthesizes the foundational methods in offline flow-based RL, categorizing them by how the energy signal is integrated and how they address computational efficiency and safety constraints.

| Algorithm | How Energy Enters | OOD Control Mechanism | Inference Cost |
|---|---|---|---|
| **EFM / QIPO (Zhang et al., 2025b)** | Modified target velocity field | Implicit (KL/Energy Target) | Moderate (ODE) |
| **FlowQ (Alles et al., 2025)** | Guided conditional path | Implicit (KL/Energy Target) | Moderate (ODE) |
| **FQL (Park et al., 2025b)** | Policy head optimization | Explicit KL-style penalty | Very Low (1-step) |
| **ReFORM (Zhang et al., 2026)** | Behavior Cloning base | Structural latent reflection | Moderate (ODE) |
| **ARFM (Zhang et al., 2025a)** | Adaptive exponential scaling | Variance-bounded weights | Moderate (ODE) |
| **SSCP (Koirala & Fleming, 2025)** | Vector completion target | Implicit (Generative prior) | Very Low (1-step) |

Table 2: A unifying comparison of foundational offline flow-matching RL algorithms.

# 5 Online Reinforcement Learning with Flow-Matching Policies

While offline RL with flow matching methods rely on weighted regression on a fixed dataset, *online RL* requires the agent to explore and update its policy through environment interaction and optimize expected return. Integrating flow matching into this setting is nontrivial because standard flow models define *deterministic* probability flow ODEs:

$$\frac{dx_t}{dt} = v_\theta(x_t, s, t), \qquad t \in [0, 1],$$

which map noise $x_0 \sim \mathcal{N}(0, I)$ to an action $a = x_1$ without inherent stochasticity during training.

Policy gradient methods, however, require stochastic policies in order to estimate gradients of $J(\theta) = \mathbb{E}_{\tau \sim \pi_\theta}[R(\tau)]$. Therefore, since in traditional RL, exploration is typically achieved by sampling from a stochastic policy (e.g., adding Gaussian noise to a mean action) but a flow policy generates actions through a multi-step Ordinary Differential Equation (ODE) solver, injecting exploration noise and calculating policy gradients is non-trivial. This fundamental mismatch has led to several distinct solutions.

**A Unified View.** Most online flow-RL methods can be understood through three design choices:

1. **How is stochasticity introduced?**

2. **How is the policy gradient computed?**

3. **How is stability ensured (KL or trust-region regularization)?**

Under this lens, existing methods fall into three representative families: (i) noise-injected stochastic flows, (ii) pathwise policy gradients through the ODE, and (iii) group-based relative optimization. We explain each in detail below.

## 5.1 Method 1: Noise Injection and Fine-Tuning

**ReinFlow** (Zhang et al., 2025c) resolves the conflict between deterministic flow-matching policies and policy gradient methods by converting deterministic ODE sampling into a discrete-time Markov process with Gaussian transitions. By injecting learnable noise into each discretized step, the method enables exact and tractable likelihood computation even with a single denoising step. The agent is optimized using a PPO-style clipped objective derived from a policy gradient formulation for Markov processes. This allows the noise network to automatically balance exploration and exploitation during training; once fine-tuning is complete, the noise module is discarded to restore the deterministic efficiency of the flow policy for deployment.

Given the Euler discretization:

$$x_{k+1} = x_k + h \, v_\theta(x_k, s, t_k),$$

ReinFlow injects learnable Gaussian noise:

$$x_{k+1} = x_k + h \, v_\theta(x_k, s, t_k) + \sigma_\phi(x_k, s, t_k) \, \xi_k, \quad \xi_k \sim \mathcal{N}(0, I).$$

This induces a stochastic policy $\pi_\theta(a|s)$ whose likelihood can be evaluated along the trajectory.

**Policy Optimization.** Because the sampling process is now stochastic, ReinFlow applies standard on-policy objectives such as PPO:

$$L^{\mathrm{PPO}}(\theta) = \mathbb{E}[\min(r_t(\theta) A_t, \mathrm{clip}(r_t(\theta), 1 - \epsilon, 1 + \epsilon) A_t)],$$

where $r_t(\theta)$ is the likelihood ratio under the stochastic flow policy.

Their algorithm works by co-training a lightweight noise injection network alongside the pre-trained flow policy's velocity field. During environment rollouts, the policy generates actions through a sequence of denoising steps where Gaussian noise is added at each stage, as defined by the noise network. This stochasticity

---

**Algorithm 6** ReinFlow (Zhang et al., 2025c) (Noise-Injected Flow RL; rough sketch)

---

1: Initialize velocity network $v_\theta$ and noise network $\sigma_\phi(\cdot)$
2: **for** each batch of environment rollouts **do**
3:     Sample $x_0 \sim \mathcal{N}(0, I)$                          $\triangleright$ $x_k$ denotes the intermediate action variable
4:     **for** $k = 0$ to $T - 1$ **do**
5:         $\xi_k \sim \mathcal{N}(0, I)$
6:         $x_{k+1} = x_k + h\, v_\theta(x_k, s, t_k) + \sigma_\phi(s, t_k)\, \xi_k$ $\triangleright$ Gaussian transitions $\Rightarrow$ tractable $\log \pi$ for PPO ratios
7:     **end for**
8:     Execute action $a = x_T$, collect reward
9: **end for**
10: Update $\theta, \phi$ via PPO using likelihood ratios from the Gaussian transitions
11: **Inference:** set $\sigma_\phi \equiv 0$ and use the deterministic ODE

---

allows the researchers to apply a derived Policy Gradient Theorem for Markov Process Policies, optimizing the agent using a clipped surrogate loss (similar to PPO) based on collected rewards. Once fine-tuning is complete, the noise network is discarded, restoring a deterministic, high-performing flow matching policy for rapid inference.

Crucially, noise is required only during training to facilitate exploration and likelihood computation. After convergence, the noise module is removed and the deterministic ODE is used for fast inference enabling efficient action generation. Thus, ReinFlow separates stochastic exploration during learning and deterministic efficiency during deployment preserving classical policy gradient machinery but modifying the generative process itself.

## 5.2 Method 2: Direct and Surrogate Policy Optimization for Flow Policies

A second family of methods avoids noise injection and instead optimizes the flow policy without adding training-time noise to the dynamics. Rather than converting the deterministic flow into a stochastic Markov chain, these approaches treat the flow policy as a reparameterized distribution whose parameters influence the final action through the ODE solution map.

Recall that a flow-matching policy generates an action $a$ by solving

$$\frac{dx_t}{dt} = v_\theta(x_t, s, t), \qquad x_0 \sim p_0, \qquad a = x_1,$$

where $v_\theta$ is a learned velocity field. The policy distribution $\pi_\theta(a|s)$ is implicitly defined by pushing forward the base distribution $p_0$ through the ODE solution operator.

The reinforcement learning objective is

$$J(\theta) = \mathbb{E}_{s \sim d^\pi, \, a \sim \pi_\theta(\cdot|s)} \left[ R(s, a) \right].$$

Since the flow model is deterministic given $x_0$, so randomness enters only through the base noise $x_0$. This enables reparameterization:

$$\nabla_\theta J(\theta) = \mathbb{E}_{x_0 \sim p_0} \left[ \nabla_a R(s, a) \frac{\partial a}{\partial \theta} \right], \qquad a = \Phi_\theta(x_0, s), \tag{19}$$

where $\Phi_\theta$ denotes the ODE solution map. Thus, gradients can be computed by differentiating through the numerical ODE solver.

**Common Algorithmic Template.** The methods in this family share a common structure that we provide in Algorithm 7. The main differences between methods lie in (i) how they estimate $\nabla_\theta \log \pi_\theta(a|s)$, (ii) whether they require explicit likelihood computation, and (iii) how they ensure numerical stability.

---

**Algorithm 7** Generic Flow-Policy Gradient Template

---

**Require:** Velocity field $v_\theta$, base distribution $p_0$
1: **for** each RL iteration **do**
2:     Sample $x_0 \sim p_0$
3:     Integrate $\dot{x}_t = v_\theta(x_t, s, t)$ to obtain $a = x_1$
4:     Execute action $a$, observe reward and advantage estimate
5:     Compute gradient of $J(\theta)$ by backpropagating through the ODE integration steps
6:     Update $\theta$ using policy gradient (or PPO-style objective)
7: **end for**

---

**Flow Matching Policy Gradients (FPO).** Flow Policy Optimization (FPO) (McAllister et al., 2025) brings flow-based generative models into the policy gradient framework without requiring exact likelihood computation. Instead of differentiating through a specific ODE sampler or evaluating the divergence of the velocity field, FPO treats the sampling procedure as a black box during environment rollouts. It constructs a surrogate likelihood ratio directly from the conditional flow matching (CFM) loss to drive a PPO-style objective. Under this formulation, maximizing the advantage-weighted ratio of the CFM losses steers the probability flow to reinforce high-return actions, making the framework agnostic to the choice of sampling method at both training and inference time.

**Reinforcement Learning for Flow-Matching Policies.** Pfrommer et al. (2025) adapt reinforcement learning to flow-matching policies in the context of variable-horizon action chunks. Their approach focuses on two RL formulations: Reward-Weighted Flow Matching (RWFM) and Group Relative Policy Optimization (GRPO).

RWFM modifies the flow-matching loss by weighting trajectories with $\exp(\alpha R)$, producing a density proportional to the reward-weighted demonstration distribution. GRPO instead samples multiple action chunks from the current flow policy, computes normalized group-relative advantages, and inserts these weights into a weighted flow-matching regression objective. Importantly, these methods do *not* differentiate through the ODE likelihood; rather, they retain the flow-matching regression structure while injecting RL signal via reward-based weighting.

**PolicyFlow: PPO for Continuous Normalizing Flows.** PolicyFlow (Yang et al., 2026) adapts Proximal Policy Optimization (PPO) to continuous normalizing flows. Computing the exact likelihood of a sample requires integrating the divergence of $v_\theta$ along the ODE trajectory:

$$\log \pi_\theta(a|s) = \log p_0(x_0) - \int_0^1 \mathrm{div}\big(v_\theta(x_t, s, t)\big)\, dt.$$

Because evaluating this integral and its gradients is computationally expensive for on-policy RL, PolicyFlow approximates the divergence term using the Hutchinson trace estimator. This yields a scalable estimate of the log-likelihood ratio needed for PPO:

$$r_t(\theta) = \exp\big(\log \pi_\theta(a_t|s_t) - \log \pi_{\theta_{\mathrm{old}}}(a_t|s_t)\big).$$

The PPO surrogate objective is then

$$L^{\mathrm{PPO}}(\theta) = \mathbb{E}\left[\min\big(r_t(\theta)A_t, \mathrm{clip}(r_t(\theta), 1-\epsilon, 1+\epsilon)A_t\big)\right].$$

This approach preserves the multi-modal expressivity of flow models while leveraging the stability guarantees of PPO.

**Key Differences.** These methods differ along three fundamental axes:

---

**Algorithm 8** Generic GRPO-style training for flow policies

---

1: **Input:** policy flow $v_\theta(x, t; s)$, prompt/state $s$, group size $G$
2: **while** training **do**
3:     **for** $i = 1, \ldots, G$ **do**
4:         Sample base noise $x_0^{(i)} \sim p_0$
5:         Generate action or output via flow sampling

$$a^{(i)} = \Phi_\theta(x_0^{(i)}, s)$$

6:         Evaluate reward $R_i = R(s, a^{(i)})$
7:     **end for**
8:     Compute group-relative advantages

$$\hat{A}_i = \frac{R_i - \text{mean}(\{R_j\})}{\text{std}(\{R_j\})}$$

9:     Update policy using PPO-style objective

$$\max_\theta \mathbb{E} \left[ \min\left( r_i(\theta)\hat{A}_i, \ \text{clip}(r_i(\theta), 1 - \epsilon, 1 + \epsilon)\hat{A}_i \right) \right]$$

10: **end while**

---

- **Gradient computation.** FPO uses a PPO-style surrogate ratio computed from conditional flow-matching losses. PolicyFlow estimates likelihood ratios via divergence approximation. RWFM/GRPO avoid likelihood computation entirely and rely on weighted regression.

- **Need for log-density.** PolicyFlow explicitly approximates $\log \pi_\theta$. FPO avoids exact log-density evaluation by using a flow matching loss based ratio proxy. RWFM/GRPO do not require density evaluation.

- **Stability mechanism.** FPO and PolicyFlow inherit PPO-style clipping, although their likelihood-ratio estimates differ. RWFM/GRPO rely on weighted regression and advantage normalization.

Conceptually, this family of methods treats the flow as a *differentiable transport map* whose parameters can be optimized either by differentiating through the ODE or by approximating policy gradients via density estimates. They contrast with noise-injection approaches (Section 5.1), which modify the generative process itself to recover tractable likelihoods.

### 5.3 Method 3: The GRPO Family (Group Relative Policy Optimization)

A rapidly growing family of methods adapts *Group Relative Policy Optimization (GRPO)* (Shao et al., 2024) to flow models. These approaches sample a group of trajectories (using different random noise seeds $x_0$), evaluate their rewards, and update the flow to increase the likelihood of the best paths.

Algorithm 8 summarizes the common structure shared by GRPO-style flow optimization methods. Instead of estimating advantages using a learned critic, these methods sample a group of candidate outputs generated from different noise seeds, evaluate their rewards, and compute normalized advantages relative to the group mean. The flow policy is then updated using a PPO-style clipped objective so that samples with higher relative reward become more likely while lower-ranked samples are suppressed. The key difference across GRPO-family methods lies in how they improve exploration, credit assignment, or variance reduction within this basic loop.

Several recent works extend this template in different ways:

- **Flow-GRPO** (Liu et al., 2025a) establishes the baseline by converting the flow ODE into an SDE to enable exploration. It demonstrates that simple group-based ranking can align large-scale text-to-image flow models with human preferences.

- **TempFlow-GRPO** (He et al., 2025) critiques the "sparse reward" assumption of standard GRPO. It argues that timing matters: decisions made early in the flow trajectory have different impacts than those made late. TempFlow introduces a temporal credit assignment mechanism to weight updates differently across flow timesteps $t \in [0, 1]$.

- **Smart-GRPO** (Yu et al., 2026) addresses the inefficiency of random exploration. Instead of sampling random noise $x_0 \sim \mathcal{N}(0, I)$, it actively optimizes the noise perturbation to explore high-reward regions, serving as a "smart" exploration strategy for the flow policy.

- **SuperFlow** (Chen et al., 2025c) tackles the variance issue. It observes that fixed group sizes in GRPO are inefficient. SuperFlow introduces variance-aware sampling that dynamically adjusts the number of samples per prompt based on the difficulty of the optimization landscape.

- **F5R-TTS** (Sun et al., 2025) applies GRPO to flow-matching based text-to-speech models. The method first reformulates the deterministic flow outputs as probabilistic Gaussian predictions, enabling likelihood-based RL updates. During fine-tuning, GRPO optimizes speech generation using rewards derived from word error rate (WER) and speaker similarity metrics, improving both intelligibility and voice consistency in zero-shot voice cloning tasks.

### 5.4 Adaptive Regularization

While reward-weighted regression provides a simple mechanism for aligning flow policies with reward signals, it can suffer from *policy collapse*, where the generative model over-optimizes the reward and converges to a degenerate distribution with low diversity. This issue is particularly severe in online fine-tuning, where the policy repeatedly samples from its own generated distribution.

To address this problem, Fan et al. (2025b) propose **Online Reward-Weighted Conditional Flow Matching with Wasserstein Regularization (ORW-CFM-W2)**, an RL-based framework for fine-tuning flow matching models with arbitrary reward functions. The core idea is to extend reward-weighted flow matching by incorporating a divergence penalty that constrains the fine-tuned model to remain close to a reference distribution.

Specifically, the induced policy update can be interpreted as updating the data distribution according to

$$q_\theta^{(n)}(x_1) \propto q_\theta^{(n-1)}(x_1) \exp\big(\lambda r(x_1) - \beta' D_{n-1}(x_1)\big),$$

where $r(x_1)$ is the reward function, $D_{n-1}(x_1)$ measures the divergence between the current model and the reference model, and $\lambda, \beta'$ control the trade-off between reward maximization and regularization. This update closely resembles classical reinforcement learning algorithms such as Advantage-Weighted Regression (AWR) (Peng et al., 2019), where policy probabilities are updated proportionally to exponentiated advantages. The divergence term acts as a stabilizer that prevents overly aggressive policy updates. Fan et al. (2025b) show that without this regularization, the policy can collapse toward a delta distribution that maximizes reward but eliminates generative diversity.

A key challenge in applying divergence regularization to continuous-time flow models is that common measures such as KL divergence are computationally expensive because evaluating likelihoods requires integrating the flow divergence along the entire ODE trajectory. To overcome this difficulty, ORW-CFM-W2 instead uses the Wasserstein-2 ($W_2$) distance as the regularization metric and derives a tractable upper bound that can be computed directly from the velocity fields of the flow model. This allows the method to control the distance between the fine-tuned model and the pre-trained reference model while avoiding expensive likelihood computations.

Algorithm 9 summarizes the backbone of ORW-CFM-W2: the model samples from its current policy online, reweights updates by reward, and uses a Wasserstein-based penalty to keep the policy close to a reference flow.

---

**Algorithm 9** Online Reward-Weighted CFM with Wasserstein Regularization (ORW-CFM-W2) (Fan et al., 2025b)

---

1: **Input:** reference flow model $v_{\theta_{\mathrm{ref}}}$, initialized policy $v_\theta \leftarrow v_{\theta_{\mathrm{ref}}}$, reward function $r(\cdot)$, weighting map $w(x_1) = F(\tau r(x_1))$, regularization coefficient $\alpha$
2: **while** not converged **do**
3:     Sample generated targets $x_1 \sim q_\theta(x_1)$ from the current flow policy
4:     Sample $t \sim \mathrm{Unif}[0,1]$ and interpolation point $x \sim p_t(x \mid x_1)$
5:     Compute reward weight $w(x_1)$ from the sampled target
6:     Compute conditional flow-matching loss:

$$\mathcal{L}_{\mathrm{CFM}} = w(x_1)\|v_\theta(t,x) - u_t(x \mid x_1)\|^2$$

7:     Compute Wasserstein regularization surrogate:

$$\mathcal{L}_{W_2} = \alpha\|v_\theta(t,x) - v_{\theta_{\mathrm{ref}}}(t,x)\|^2$$

8:     Update $\theta$ by minimizing:

$$\mathcal{L}_{\mathrm{ORW\text{-}CFM\text{-}W2}} = \mathcal{L}_{\mathrm{CFM}} + \mathcal{L}_{W_2}$$

9: **end while**
10: **Return:** fine-tuned flow policy $v_\theta$

---

Table 3: Comparison of online reward-weighted flow fine-tuning methods.

| Method | Regularization | Sampling | Key Idea |
|---|---|---|---|
| ORW-CFM-W2 | Fixed $W_2$ penalty | Online | Reward-weighted flow matching |
| ADRPO | Adaptive $W_2$ penalty | Online | Advantage-dependent regularization |

Compared with offline reward-weighted regression, the crucial difference is that the sampling distribution is updated online from the current policy itself, which improves exploration but also makes regularization essential for preventing collapse.

While ORW-CFM-W2 stabilizes training through a fixed Wasserstein regularization coefficient, choosing a single global penalty introduces a fundamental trade-off: strong regularization preserves diversity but slows reward optimization, while weak regularization enables learning but risks policy collapse. To address this limitation, Fan et al. (2025c) propose **Adaptive Divergence Regularized Policy Optimization (ADRPO)**, which dynamically adjusts the regularization strength using the advantage signal. Intuitively, samples with high advantage are allowed to deviate more from the reference policy, while samples with uncertain or low advantage are constrained more strongly. This adaptive mechanism allows the policy to aggressively exploit reliable reward signals while remaining conservative when the optimization signal is noisy. Algorithm 10 summarizes this adaptive variant.

Conceptually, ADRPO generalizes ORW-CFM-W2 by replacing the fixed divergence penalty with an advantage-dependent one. Whereas ORW-CFM-W2 applies the same regularization strength to all samples, ADRPO adapts the constraint to the local reward landscape. This improves the exploration-exploitation trade-off: the policy can move more freely toward reliably high-reward regions while remaining anchored to the reference model when the reward signal is ambiguous. Empirically, the authors show that this adaptive regularization improves alignment quality while maintaining diversity in generative outputs.

In practice, ADRPO can be integrated with existing RL fine-tuning frameworks, including GRPO or reward-weighted flow matching, by replacing the fixed divergence penalty with an advantage-dependent coefficient. This simple modification significantly improves the reward-diversity trade-off during training and has been shown to outperform fixed-regularization methods such as ORW-CFM-W2 in tasks such as text-to-image alignment and multimodal reasoning.

---

**Algorithm 10** Adaptive Divergence Regularized Policy Optimization (ADRPO) (Fan et al., 2025c)

---

1: **Input:** reference flow model $v_{\theta_{\text{ref}}}$, initialized policy $v_\theta \leftarrow v_{\theta_{\text{ref}}}$, reward function $r(\cdot)$, weighting function $F(\cdot)$
2: **while** not converged **do**
3:      Sample generated targets $x_1 \sim q_\theta(x_1)$ from the current flow policy
4:      Compute reward $r(x_1)$ and advantage estimate $A(x_1)$
5:      Sample $t \sim \text{Unif}[0,1]$ and interpolation point $x \sim p_t(x \mid x_1)$
6:      Compute reward weight
$$w(x_1) = F(\tau A(x_1))$$
7:      Compute adaptive regularization coefficient

$$\alpha(x_1) = g(A(x_1))$$

where $g(\cdot)$ decreases the penalty for high-advantage samples.
8:      Compute weighted flow matching loss

$$\mathcal{L}_{\text{CFM}} = w(x_1) \, \|v_\theta(t,x) - u_t(x|x_1)\|^2$$

9:      Compute adaptive Wasserstein regularization

$$\mathcal{L}_{W_2} = \alpha(x_1) \, \|v_\theta(t,x) - v_{\theta_{\text{ref}}}(t,x)\|^2$$

10:      Update $\theta$ by minimizing
$$\mathcal{L}_{\text{ADRPO}} = \mathcal{L}_{\text{CFM}} + \mathcal{L}_{W_2}$$

11: **end while**
12: **Return:** fine-tuned flow policy $v_\theta$

---

## 5.5 Bridging Diffusion and Flows: Forward Process Optimization

While this survey focuses on flow matching, boundary-crossing methods provide valuable insights into online optimization. For instance, **DiffusionNFT** (Zheng et al., 2025b) tackles the intractable likelihoods of online diffusion RL by optimizing the model directly on the forward process. By contrasting positive and negative generations to define an implicit policy improvement direction, DiffusionNFT translates the reinforcement signal into a supervised flow-matching objective. This demonstrates how flow-matching mechanics can resolve fundamental optimization bottlenecks even in architectures originally designed around diffusion.

# 6 Beyond Policies: Critics, Dynamics, and Alignment

Flow matching has demonstrated utility beyond acting as a policy, finding applications in critic modeling, dynamics adaptation, and complex reasoning tasks.

## 6.1 floq: Training Critics via Flow Matching

While most flow-based reinforcement learning methods focus on improving the *policy*, the critic is typically still implemented as a conventional neural network regressor that maps $(s,a)$ directly to a scalar value $Q(s,a)$. **floq** (Agrawalla et al., 2025) proposes a fundamentally different perspective: instead of treating the Q-function as a static mapping, it parameterizes the critic itself as a *flow model*.

Concretely, floq models the Q-value as the output of a continuous transformation of a latent scalar variable $z$ governed by a velocity field. Starting from an initial sample $z_0$ drawn from a simple base distribution (e.g., $z_0 \sim \text{Unif}[l,u]$), a time-dependent velocity field $v_\theta(t,z \mid s,a)$ transports the sample toward the target Q-value through an ODE integration process. After $K$ integration steps, the final value $z_K$ represents the

---

**Algorithm 11** Flow-Matching Q-Function Training (floq) (Agrawalla et al., 2025)

---

1: **Input:** dataset $\mathcal{D}$, flow critic parameters $\theta$, policy $\pi_\phi$, integration steps $K$
2: **while** not converged **do**
3:     Sample transition $(s, a, r, s') \sim \mathcal{D}$
4:     Sample latent value $z_0 \sim \text{Unif}[l, u]$
                                                                             $\triangleright$ Flow critic forward pass
5:     **for** $k = 0, 1, \ldots, K - 1$ **do**
6:         $z_{k+1} \leftarrow z_k + \frac{1}{K} v_\theta(k/K, z_k \mid s, a)$
7:     **end for**
8:     $Q_\theta(s, a) \leftarrow z_K$
                                                              $\triangleright$ Temporal-difference target
9:     Sample $a' \sim \pi_\phi(\cdot | s')$
10:    $y \leftarrow r + \gamma Q_{\theta^-}(s', a')$
                                                               $\triangleright$ Update velocity field
11:    Update $\theta$ using TD loss:
$$\mathcal{L} = (Q_\theta(s, a) - y)^2$$
12: **end while**
13: **Return:** flow critic $Q_\theta$

---

predicted Q-value:

$$z_{k+1} = z_k + \frac{1}{K} v_\theta\left(\frac{k}{K}, z_k \mid s, a\right), \qquad k = 0, \ldots, K - 1,$$

with the critic prediction given by

$$Q_\theta(s, a) = z_K.$$

Intuitively, this formulation views value estimation as a *flow process that progressively refines an initial guess into the correct Q-value*. The velocity field is trained using a temporal-difference objective so that the terminal distribution of $z_K$ concentrates around the true value $Q^\pi(s, a)$. Importantly, because the Q-value is produced through iterative numerical integration, the number of integration steps $K$ acts as a controllable compute parameter.

This parameterization introduces a useful property absent from conventional critics: *test-time compute scaling*. Increasing the number of integration steps $K$ effectively increases the depth of the computation used to estimate the value function. As a result, floq can improve value estimation accuracy by allocating additional inference compute, similar to iterative reasoning in modern generative models. Empirically, Agrawalla et al. (2025) show that increasing the number of flow steps consistently improves performance on challenging offline RL benchmarks, outperforming critics based on standard monolithic neural networks.

Conceptually, floq differs from earlier flow-based RL methods in that it applies flow matching to the *critic* rather than the policy. Most prior work leverages flows to model expressive action distributions, while leaving the value function unchanged. floq instead introduces iterative computation into value estimation itself. This approach provides a new scaling axis for reinforcement learning: increasing sequential computation in the critic, rather than increasing network width or ensemble size. Although floq focuses on value-based RL, the idea of applying flow models to value estimation opens an interesting research direction for integrating generative modeling with critic learning.

### 6.2 Dynamics and Transfer

While most flow-based RL work focuses on learning expressive policies or critics, another emerging direction uses flow matching to address *transfer learning under dynamics shift*. In many practical scenarios, an agent has access to an offline dataset collected in a *source environment*, but must operate in a *target environment* whose transition dynamics differ. Formally, the source and target Markov decision processes share the same

---

**Algorithm 12** Composite Flow Matching for Dynamics Transfer (CompFlow) (Kong et al., 2025)

---

1: **Input:** offline dataset $\mathcal{D}_{\mathrm{src}}$, target environment
2: Train offline flow model $v_\theta^{\mathrm{off}}$ to model source transitions
3: **while** interacting with target environment **do**
4:     Collect target transition $(s, a, s')$
5:     Initialize latent sample $x_0 \sim p_0$
                                                                                 ▷ Offline dynamics flow
6:     $x^{\mathrm{src}} \leftarrow \mathrm{Flow}_\theta^{\mathrm{off}}(x_0, s, a)$
                                                                                   ▷ Online adaptation flow
7:     $x^{\mathrm{tgt}} \leftarrow \mathrm{Flow}_\phi^{\mathrm{on}}(x^{\mathrm{src}}, s, a)$
8:     Estimate dynamics discrepancy between source and target flows
9:     Use discrepancy signal to guide exploration and policy updates
10: **end while**
11: **Output:** policy adapted to target dynamics

---

state and action spaces but differ in transition dynamics:

$$P_{\mathrm{src}}(s'|s, a) \neq P_{\mathrm{tgt}}(s'|s, a).$$

This mismatch, often referred to as the *dynamics gap*, can significantly degrade performance if offline data are used directly for policy learning.

**Composite Flow Matching (CompFlow)** (Kong et al., 2025) proposes a principled approach for estimating and exploiting this dynamics gap using flow matching and optimal transport. The key idea is to model the transition distributions of the source and target environments using flow-based generative models and compute a transport map that moves samples from the source transition distribution toward the target distribution. Because flow matching has a natural connection to optimal transport, the magnitude of the learned transport provides a measure of distributional discrepancy between the two dynamics models. Specifically, the discrepancy can be quantified using the Wasserstein distance between the two transition distributions:

$$W_2^2(P_{\mathrm{src}}, P_{\mathrm{tgt}}) = \inf_{q \in \Pi(P_{\mathrm{src}}, P_{\mathrm{tgt}})} \mathbb{E}_{(x_0, x_1) \sim q} \left[ \|x_0 - x_1\|_2^2 \right],$$

where $\Pi(P_{\mathrm{src}}, P_{\mathrm{tgt}})$ denotes the set of couplings between the two transition distributions. In the flow-matching formulation, the learned transport map approximates the optimal transport plan between the two distributions, enabling the Wasserstein distance to be estimated from the flow trajectories.

**Composite Flow Architecture.** To estimate this dynamics gap robustly, CompFlow constructs a *composite flow model* consisting of two stages. First, an offline flow model learns the transition distribution from the source dataset:

$$\frac{d}{dt}\psi_\theta^{\mathrm{off}}(x_0, t|s, a) = v_\theta^{\mathrm{off}}\big(\psi_\theta^{\mathrm{off}}(x_0, t|s, a), t, s, a\big).$$

This flow transports latent noise $x_0$ into a representation of the source transition distribution. A second flow model then adapts this representation toward the target transition distribution observed during online interaction:

$$\frac{d}{dt}\psi_\phi^{\mathrm{on}}(x, t|s, a) = v_\phi^{\mathrm{on}}\big(\psi_\phi^{\mathrm{on}}(x, t|s, a), t, s, a\big).$$

By initializing the online flow from the output of the offline flow rather than from a Gaussian prior, the model effectively learns a transport map from the source dynamics to the target dynamics. This composite structure improves generalization when only limited online data are available. The overall procedure of CompFlow is summarized in Algorithm 12.

**Using the Dynamics Gap for Exploration.** Once the discrepancy between transition distributions is estimated, CompFlow leverages it in two ways. First, offline transitions that exhibit *low dynamics gap*

---

**Algorithm 13** Preference Flow Matching (PFM) (Kim et al., 2024)

---

1: **Input:** preference dataset $\mathcal{D} = \{(x^+, x^-)\}$
2: **while** training **do**
3:     Sample preference pair $(x^+, x^-)$
4:     Sample interpolation time $t \sim \text{Uniform}(0, 1)$
5:     Construct interpolation
$$x_t = (1 - t)x^- + tx^+$$
6:     Compute target velocity
$$u_t = x^+ - x^-$$
7:     Update flow model by minimizing
$$\|v_\theta(x_t, t) - u_t\|^2$$
8: **end while**

---

relative to the online environment are preferentially incorporated into policy training, improving sample efficiency. Second, the estimated discrepancy is used as an exploration signal: the agent is encouraged to collect new data in regions where the dynamics gap is large, which correspond to parts of the state-action space that are poorly represented in the offline dataset.

CompFlow demonstrates that flow matching can be used not only for policy generation but also as a principled tool for *distributional discrepancy estimation*. By leveraging the optimal transport interpretation of flow models, the method provides a geometrically meaningful measure of dynamics mismatch and uses it to guide both data selection and exploration in transfer reinforcement learning.

### 6.3 Preference Alignment and Reasoning

Flow matching is increasingly explored as an alternative to diffusion models for preference alignment tasks, particularly in reinforcement learning from human feedback (RLHF) and reasoning optimization.

**Preference Flow Matching (PFM)** (Kim et al., 2024) proposes a principled method for aligning generative models directly from preference pairs without explicitly training a reward model. Given a preferred sample $x^+$ and a rejected sample $x^-$, the method constructs a *preference transport* objective that encourages the model to transform rejected outputs toward preferred ones along a learned flow trajectory. Instead of optimizing a scalar reward, PFM modifies the flow-matching regression target so that the velocity field points from the rejected sample toward the preferred sample, effectively learning a transport map between preference distributions.

Formally, the model learns a velocity field $v_\theta(x_t, t)$ along interpolation paths between rejected and preferred samples:

$$x_t = (1 - t)x^- + tx^+, \tag{20}$$

with target velocity

$$u_t = x^+ - x^-. \tag{21}$$

The model is trained via a regression objective similar to standard flow matching, but using preference-induced transport targets:

$$\mathcal{L}_{\text{PFM}} = \mathbb{E}\left[\|v_\theta(x_t, t) - u_t\|^2\right]. \tag{22}$$

Algorithm 13 summarizes the training procedure.

Beyond direct preference alignment, flow-based objectives have also been explored for reasoning optimization. **FlowRL** (Zhu et al., 2025) proposes matching the distribution of high-reward reasoning trajectories rather than maximizing scalar rewards. This approach, termed *Flow Balancing*, encourages the model to approximate the full distribution of successful reasoning paths, preserving diversity while improving correctness.

Overall, these works illustrate how flow matching can provide an alternative alignment mechanism that operates directly on distributions of preferred outputs, avoiding explicit reward modeling and potentially offering more stable training dynamics for large generative models.

### 6.4 General Guidance Frameworks

While many flow-based RL algorithms introduce guidance heuristically (e.g., weighting losses with $Q$-values or adding gradients from value functions), recent work has begun to formalize a general theory of *guided flow matching*. Feng et al. (2025a) propose the first unified framework for guidance in flow-based generative models.

The central idea is to reinterpret guidance as modifying the *velocity field* that defines the probability transport process. Recall that a flow model generates samples by solving the ODE

$$\frac{dx_t}{dt} = v_t(x_t), \tag{23}$$

which transports samples from a base distribution $p_0$ to a target distribution $p_1$.

Suppose we wish to guide generation using an *energy function* $J(x)$ (for example, a classifier score, reward function, or value estimate). The desired guided distribution is

$$p'(x) \propto p(x) \exp(-J(x)), \tag{24}$$

which increases the probability of samples with lower energy values.

The key theoretical question is therefore:

> *How should we modify the original vector field $v_t(x)$ so that the resulting flow generates samples from the new distribution $p'(x)$?*

The framework shows that guidance can be implemented by adding a *guidance vector field* $g_t(x)$ to the original velocity field:

$$v'_t(x) = v_t(x) + g_t(x), \tag{25}$$

where the modified vector field $v'_t$ transports samples toward the guided distribution $p'(x)$.

Under this formulation, many previously proposed methods can be interpreted as different approximations of the optimal guidance field $g_t(x)$. The paper derives several families of guidance strategies:

- **Monte Carlo guidance** ($g_{\text{MC}}$)**:** an asymptotically exact estimator of the optimal guidance field obtained via sampling.

- **Gradient-based guidance** ($g_{\text{local}}$)**:** an approximation proportional to the gradient of the energy function, similar to classifier or value guidance.

- **Covariance-preconditioned guidance** ($g_{\text{cov}}$)**:** a refined form that scales gradients using the covariance of the conditional distribution along the flow.

- **Training-based guidance** ($g_\phi$)**:** methods that learn the guidance vector field with an auxiliary neural network.

Importantly, the authors show that classical techniques such as classifier guidance, energy guidance, and several diffusion-based guidance mechanisms emerge as special cases of this general flow-guidance framework under additional assumptions about the probability path.

From the perspective of reinforcement learning, this theory provides a principled interpretation of many existing algorithms. For example, Q-guided flows and energy-weighted flow matching can be viewed as

approximations to the optimal guidance vector field where the energy function corresponds to the negative value function (i.e., $J(x) = -Q(x)$).

Thus, the guided-flow framework provides a unifying lens through which diverse RL algorithms ranging from reward-weighted regression to gradient-guided sampling can be interpreted as modifying the underlying transport dynamics of the flow model.

These developments suggest that flow matching provides a broader framework for integrating guidance signals into generative models, extending beyond policy learning to value estimation, transfer learning, and alignment.

**Discussion and Unifying Perspective.** Taken together, the works reviewed in this section illustrate that flow matching is not limited to modeling action policies, but can serve as a general computational primitive for reinforcement learning. For instance, **floq** extends flow matching to *value estimation* by parameterizing the critic itself as a flow process, enabling test-time compute scaling for value-based RL. Similarly, **Composite Flow Matching** (CompFlow) leverages transport maps to address *dynamics shifts* in transfer learning, treating environment mismatch as a probability transport problem between transition distributions. In the context of model alignment, methods such as **Preference Flow Matching** reinterpret preference learning as transporting rejected outputs toward preferred ones, while recent work on **guided flows** provides a general theoretical framework for incorporating external signals such as rewards, classifiers, or value functions into the flow dynamics. Viewed through this lens, many seemingly disparate algorithms can be understood as modifying or learning vector fields that reshape probability transport toward desirable regions of the solution space. This perspective suggests a broader research direction in which flow-based transport processes unify policy learning, value estimation, preference alignment, and environment adaptation within a common generative modeling framework.

# 7 Emerging Frontiers: Action Spaces, Architectures, Value Estimation, and Alignment

As flow matching matures within reinforcement learning, researchers are rapidly extending its core principles beyond standard benchmarks. Recent literature discussed here, which to our knowledge has not yet been peer-reviewed at the time of writing this survey, has focused on scaling flow models to complex action spaces, restructuring the ODE architectures to stabilize off-policy optimization, applying flow matching to value estimation itself, and refining the alignment of large-scale foundation models. This section surveys these emerging frontiers.

## 7.1 Structured and Non-Euclidean Action Spaces

Standard flow matching assumes continuous, Euclidean action spaces. When actions are discrete, combinatorially constrained, or defined on curved manifolds, the velocity fields, spatial gradients, and linear interpolation that underlie conditional flow matching are either undefined or geometrically invalid. Recent work addresses this through a shared design logic by replacing the continuous ODE with a transport process that respects the target geometry while preserving simulation-free training.

For finite action sets, Khan et al. (2026) utilize the discrete flow matching framework (Campbell et al., 2024; Gat et al., 2024; Lipman et al., 2024) and formulate it for RL with discrete actions. Specifically, they replace the continuous ODE with a Continuous-Time Markov Chain (CTMC) that transports probability via discrete jumps governed by the Kolmogorov forward equation. Khan et al. (2026) then extend the approach to multi-agent settings through factorized per-agent rate functions that reduce the joint action space from $\prod_i |\mathcal{A}_i|$ to $\sum_i |\mathcal{A}_i|$. For hard combinatorial constraints, Kong et al. (2026) retain a continuous flow but place it on a latent unit sphere, delegating feasibility to a downstream combinatorial optimization solver and stabilizing the resulting discontinuous value landscape with a smoothed Bellman operator. For robotic end-effector poses on $SE(3)$ (the group of 3D rigid-body translations and rotations), Braun et al. (2024); Ding et al. (2025) replace linear interpolation with geodesic paths under the Riemannian metric, with the

latter work applying LaSalle's invariance principle to guarantee stable convergence to the target manifold distribution.

These methods enforce feasibility through complementary mechanisms: algebraic constraints in the discrete case (jumps cannot leave the action set), optimization-based projection in the combinatorial case (solvers return feasible outputs by construction), and geometric constraints in the Riemannian case (geodesic flows preserve the manifold). A notable asymmetry, however, is that the discrete and combinatorial frameworks integrate Q-weighted policy improvement through energy-guided regression, while the Riemannian policies have so far been evaluated only in imitation learning without any value or reward signal. Bridging this gap, and combining multiple types of geometric structure within a single framework, remain open directions.

## 7.2 Architectural Innovations for Stability and Acceleration

Optimizing flow policies via online RL requires backpropagating through the multi-step ODE solver, which produces the same vanishing and exploding gradient pathologies that afflict deep recurrent networks. This shared bottleneck has prompted three converging but distinct strategies, each trading off expressiveness, inference cost, and implementation complexity in different ways.

**Redesigning the ODE backbone.** The most direct response is to stabilize the gradient path through the flow rollout itself. Zhang et al. (2025d) formalize this connection by proving that the Euler discretization of a flow policy is algebraically equivalent to a residual RNN computation, then reparameterize the velocity network with GRU-style gating and Transformer-style cross-attention to regulate gradient flow across sampling steps. This enables direct end-to-end off-policy training via SAC without policy distillation or surrogate objectives. From a complementary angle, Li et al. (2026) address the same instability at the objective level rather than the architecture level, reformulating the training target as a posterior mean estimation problem and using Langevin Stein operators to construct minimum-variance control variates. Both approaches retain the full multi-step flow and its multimodal expressiveness but introduce additional computational or architectural overhead during training.

**Freezing the flow and learning a correction.** An alternative strategy avoids differentiating through the flow entirely by freezing a pretrained flow model and learning a lightweight additive policy on top. Dong et al. (2025) train a small Gaussian "edit policy" that shifts each action sampled from the frozen base toward higher Q-value regions, with a non-parametric selection step that chooses the best among original and edited candidates. Su et al. (2026) apply a similar residual steering approach to dexterous manipulation, and Wan et al. (2025) adopt a related perspective where a simple student MLP explores the environment guided by an expressive flow-based reward model. These methods are straightforward to implement and avoid gradient instability by construction, but they restrict policy improvement to local refinements around the base distribution and cannot discover entirely new modes that the pretrained flow does not cover.

**Collapsing the ODE to a single step.** The most aggressive strategy eliminates multi-step inference altogether. Chen et al. (2025d) show that the one-step sampling error of a straight-interpolation flow is bounded by the variance of the target distribution, which shrinks as the policy converges, making single-step generation increasingly accurate over the course of training. Nguyen & Yoo (2025) and Li et al. (2025) take a different route by training average velocity fields (MeanFlow (Geng et al., 2026)) that directly map noise to actions in a single function evaluation without iterative integration. The latter extends this idea to decentralized multi-agent offline RL, where the cost of multi-step sampling is compounded across agents. Single-step methods achieve inference speeds matching standard Gaussian policies but may sacrifice multimodal expressiveness when the target distribution retains high variance early in training.

These three strategies form a spectrum. Backbone redesign preserves the full generative capacity of the flow but requires non-trivial architectural changes and does not reduce inference cost. Residual correction is the simplest to deploy on top of any pretrained model and inherits the base flow's expressiveness for free, but cannot move far from the prior distribution. Single-step collapse achieves the fastest inference and is particularly attractive for real-time robotics, but its quality guarantee depends on the target distribution having low variance, a condition that holds at convergence but not during early exploration. A natural

but unexplored direction is to combine these ideas, for example by using a residual correction during early training when multimodality matters and transitioning to single-step generation as the policy converges.

## 7.3 Generative Value Estimation (Distributional RL)

While most flow-based RL methods focus on making the actor more expressive, the critic typically remains a conventional neural network that maps state-action pairs to a single scalar $Q(s, a)$. Recent work argues that this creates an information bottleneck: a highly expressive actor receives only a low-dimensional learning signal from a simple critic, which cannot capture multimodality, skewness, or heavy tails in the return distribution. A growing line of research addresses this asymmetry by applying flow matching to the critic itself, modeling the full distribution of returns rather than just its expectation.

The core idea shared across these methods is to parameterize the Q-function as a conditional flow model that transports samples from a base distribution to the distribution of discounted returns, trained via a distributional Bellman target. Agrawalla et al. (2025) introduced this perspective with floq (discussed in detail in Section 6.1), which models value estimation as an iterative flow process over a scalar latent variable and showed that increasing the number of integration steps at test time consistently improves value accuracy, providing a new "test-time compute" scaling axis for critics. Chen et al. (2025a) extend this idea with a two-stage Distributional Flow Critic (DFC) that trains a target flow network via flow matching on Bellman return samples and then distills its output into a one-step quantile critic via Wasserstein regression, avoiding the need to backpropagate through an ODE solver during actor updates. Zhong et al. (2025) propose FlowCritic with a related architecture that bridges flow-based value generation with efficient gradient computation for the actor. Espinosa-Dice et al. (2025) take the idea furthest with EVOR, which learns the full reward-to-go distribution via a flow-based temporal difference objective and demonstrates that pairing expressive flow actors with expressive flow critics enables scalable offline RL without the compounding errors introduced by policy distillation.

A recurring tension across these methods is computational. Generating return samples from the critic requires solving an ODE, which introduces the same backpropagation through time instability that motivated one-step distillation on the actor side. The solutions mirror those from the actor literature: floq embraces the multi-step computation as a feature, DFC distills the flow critic into a cheaper one-step proxy, and EVOR avoids the issue by using the flow critic only for target construction with stop-gradient operators. Whether distributional flow critics can be combined with online RL objectives, where the return distribution shifts continuously during training, remains largely untested.

## 7.4 Fine-Grained Alignment and Trajectory Planning

As flow matching scales from continuous control benchmarks to foundation models and long-horizon planning, two related challenges emerge. First, RL fine tuning of large pretrained flow models risks destroying the learned prior if the reward signal is noisy or the optimization is too aggressive. Second, sparse terminal rewards provide no signal about which decisions along the flow trajectory actually mattered. Recent work addresses both problems, and the solutions are converging on a shared insight that the internal structure of the flow generation process itself can be exploited for more granular control.

**Preserving the pretrained prior.** When flow matching is used as the action decoder in Vision-Language-Action (VLA) models such as $\pi_0$, standard policy gradient methods are infeasible because exact likelihood ratios require solving probability-flow ODEs with Jacobian trace terms. Three recent methods propose different workarounds that all aim to keep the RL update aligned with the original flow matching objective. Lyu et al. (2025) construct a likelihood-free policy ratio from per sample changes in the conditional flow matching loss, enabling PPO-style clipped updates without density evaluation. Xue et al. (2025) take a more direct route by proving that DDPO-style per step likelihood optimization is equivalent to denoising score matching with noisy targets, which increases variance, and propose Advantage Weighted Matching (AWM), which simply reweights the original flow matching loss by exponentiated advantages to achieve up to $24\times$ faster convergence than Flow-GRPO. Wang et al. (2025b) diagnose a subtler failure mode in which the importance-ratio distribution in Flow-GRPO is systematically left shifted and varies across timesteps,

causing the clipping mechanism to fail for positive advantage samples and driving implicit over optimization. Their GRPO-Guard method restores balanced clipping through ratio normalization and gradient reweighting across denoising steps. Together these methods illustrate that naive transfer of LLM-style GRPO to flow models introduces biases that require flow-specific corrections.

**Intermediate credit assignment.** Standard GRPO assigns a single terminal reward to the entire flow trajectory, ignoring the fact that early high noise steps have qualitatively different exploratory impact than late refinement steps. Zhou et al. (2025) and Fan et al. (2025a) introduce mechanisms to evaluate intermediate states along the generation path, effectively solving the temporal credit assignment problem within the flow itself. Fine-Grained GRPO uses per step reward signals derived from partial generations, while AC-Flow provides intermediate feedback through auxiliary evaluators at selected flow timesteps. These approaches complement the temporal reweighting ideas in TempFlow-GRPO (discussed in Section 5.3) and point toward a broader principle that the flow trajectory should be treated as a sequential decision process with its own internal reward structure.

**Trajectory-level generation.** A parallel line of work moves beyond single action generation to model temporally extended trajectories as the output of the flow. This reframes planning as conditional generation, where the flow transports noise to complete state-action sequences rather than individual actions. Feng et al. (2025b) learn the full ODE solution map for entire trajectories, while Diao et al. (2025) and Rouxel et al. (2025) adapt flow matching specifically for offline goal-conditioned RL using hindsight relabeling and extremum seeking objectives respectively. Zheng et al. (2025a) take a different approach by modeling future state occupancy distributions rather than action sequences, learning intention conditioned flow occupancy models that predict which states an agent will visit under different latent task intentions. This enables pre-training on heterogeneous multi-task datasets and efficient fine-tuning via generalized policy improvement. Wang et al. (2025a) constrain the generated trajectories using non-linear Controllability Gramian matrices, ensuring that planned paths remain dynamically feasible. These trajectory level methods share the advantage of capturing long range temporal dependencies that single step action policies cannot represent, but they face the challenge that trajectory distributions are higher dimensional and harder to model accurately, and most have been evaluated only in relatively simple environments.

**Summary.** The four frontiers surveyed in this section all extend the same underlying principle, namely learning a velocity field that transports probability mass toward desirable outcomes, but they do so along orthogonal axes. Structured action spaces generalize the transport mechanism beyond Euclidean geometry. Architectural innovations address the gradient and latency bottlenecks that arise when the ODE backbone meets online RL objectives. Distributional critics apply flow matching to the value function rather than the policy, providing richer learning signals. And alignment and trajectory planning methods exploit the internal temporal structure of the flow itself, either to assign credit at intermediate generation steps or to model entire trajectories as the unit of generation. A recurring observation across all four areas is that ideas initially developed for one setting tend to transfer to others. Single-step acceleration techniques from the architecture literature are now appearing in distributional critics and trajectory planners. Energy-weighted regression from offline policy learning is being adapted to discrete and combinatorial spaces. And the temporal credit assignment mechanisms designed for foundation model alignment are closely related to the process reward ideas emerging in trajectory-level generation. These cross-cutting connections suggest that flow matching is maturing into a unified computational substrate for reinforcement learning, though significant gaps remain in combining multiple extensions within a single framework.

## 8  Applications of Flow Matching in RL

While the preceding sections were organized around algorithmic principles, this section synthesizes how flow-matching RL methods are being deployed across application domains. Many of the algorithms cited above are not domain agnostic in practice, since their design choices (action-space geometry, inference budget, data modality) are typically driven by a target application.

**Robotics and visuomotor control.** Robotics is the largest application area for flow matching policies. Diffusion Policy and 3D Diffusion Policy popularized expressive visuomotor policies on real robot manipulation (Chi et al., 2023; Ze et al., 2024; Ma et al., 2024), and the flow matching successors target the inference latency bottleneck that limits diffusion models in high frequency control. SSCP achieves single step generation (Koirala & Fleming, 2025), FQL deploys a one step distilled actor with a flow prior (Park et al., 2025b), and the Riemannian Flow Matching Policies (RFMP/SRFMP) handle non-Euclidean end-effector states on $SE(3)$ (Braun et al., 2024; Ding et al., 2025). For Vision-Language-Action (VLA) models, FPO fine tunes large pre-trained flow policies such as $\pi_0$ (Lyu et al., 2025), while ARFM addresses signal/variance trade-offs for VLA-scale post-training (Zhang et al., 2025a). Residual policies such as EXPO (Dong et al., 2025) and RFS (Su et al., 2026) support local refinement on dexterous manipulation without disturbing the multimodal flow prior.

**Image and video generation alignment.** Flow matching has rapidly become the backbone of choice for RL based fine tuning of large generative models. The GRPO family (Flow-GRPO, TempFlow-GRPO, Smart-GRPO, SuperFlow) is primarily evaluated on text-to-image alignment with human preference reward models such as PickScore or ImageReward (Liu et al., 2025a; He et al., 2025; Yu et al., 2026; Chen et al., 2025c), while GRPO-Guard, AWM, and Fine-Grained GRPO target stability and credit assignment for the same setting (Wang et al., 2025b; Xue et al., 2025; Zhou et al., 2025). DiffusionNFT applies flow matching to online RL on diffusion backbones for image generation (Zheng et al., 2025b).

**Speech and audio.** F5R-TTS applies GRPO to a flow matching text-to-speech model, optimizing word error rate and speaker similarity rewards for zero-shot voice cloning (Sun et al., 2025), and YingMusic-SVC extends Flow-GRPO with singing specific inductive biases for zero-shot singing voice conversion (Chen et al., 2025b).

**Language model reasoning.** FlowRL reframes RL fine tuning of LLM reasoning chains as matching a target *distribution* of high reward trajectories rather than maximizing a scalar reward, preserving solution diversity in mathematical reasoning benchmarks (Zhu et al., 2025).

**Autonomous driving and traffic simulation.** Although most diffusion-based driving and traffic-simulation systems (Zheng et al., 2025c; Zhong et al., 2023) have not yet been re-implemented in pure flow matching form, they share the planning as generation framing that flow matching trajectory methods such as GTP (Feng et al., 2025b) and CtrlFlow (Wang et al., 2025a) build on, suggesting that the planning as generation framing may transfer naturally to deterministic flow based planners.

**Multi-agent coordination and goal-conditioned navigation.** MAC-Flow (Lee et al., 2025a) models centralized joint action distributions and distills them into fast decentralized one step policies, demonstrating multi-agent coordination benchmarks beyond standard continuous control. Goal-conditioned trajectory generation (Diao et al., 2025) and Extremum Flow Matching (Rouxel et al., 2025) apply flow matching to humanoid and robotic goal reaching, while Intention-Conditioned Flow Occupancy Models predict future state visitation for navigation type tasks (Zheng et al., 2025a).

**Transfer learning under dynamics shift.** CompFlow uses flow matching as a distributional discrepancy estimator between source and target environment dynamics, leveraging the optimal transport interpretation of flow matching to guide both data selection and exploration (Kong et al., 2025).

## 9 Benchmarks and Evaluation Protocols

Because the literature on flow matching in reinforcement learning spans offline RL, online RL, planning, and model alignment, papers are often evaluated under different protocols. As a result, empirical comparisons (provided in Section 10) should be interpreted with care: improvements reported on one benchmark or under one inference budget do not always transfer directly to another setting. For this reason, it is useful to briefly summarize the most common benchmark families and evaluation metrics before comparing Gaussian, diffusion, and flow-based policies.

**Offline RL Benchmarks.** A large fraction of the literature reviewed in this survey evaluates methods on **D4RL** (Fu et al., 2020), which remains the standard benchmark suite for offline continuous control RL. D4RL contains domains such as Gym locomotion, AntMaze, and Adroit, and typically reports *normalized return*, allowing results from different tasks to be compared on a common scale. Representative diffusion-based and flow-based offline RL works including Diffusion-QL (Wang et al., 2023), IDQL (Hansen-Estruch et al., 2023), Energy-Weighted Flow Matching / QIPO (Zhang et al., 2025b), and several efficiency-oriented flow variants, all report D4RL results.

More recent flow based offline RL work has also adopted **OGBench** (Park et al., 2025a), a broader benchmark designed to evaluate standard reward maximizing offline RL across state based and pixel based tasks. Flow Q-Learning (FQL) (Park et al., 2025b) uses OGBench as its main benchmark while also reporting results on D4RL. Table 5 in Section 10 collects reported results across these benchmarks for Gaussian, diffusion, and flow based offline RL policies.

**Online RL and Continuous Control Benchmarks.** For online policy optimization, papers typically report average episodic return on standard continuous control environments such as the **Gymnasium MuJoCo** suite (Todorov et al., 2012; Foundation, 2023), the **MuJoCo Playground** (McAllister et al., 2025), or the **DeepMind Control Suite** (Tassa et al., 2018). In this setting, evaluation focuses not only on final return but also on *sample efficiency*, i.e., how quickly performance improves with environment interaction. This distinction is especially important in online FM-based RL, because some methods optimize expressive policies directly through the ODE, while others introduce stochasticity or group based updates to make policy optimization tractable.

Furthermore, a practical consequence of the rapid growth in this area is that flow matching online RL papers have not yet converged on a shared benchmark suite. FPO (McAllister et al., 2025) and PolicyFlow (Yang et al., 2026) are evaluated on MuJoCo Playground and the DeepMind Control Suite, ReinFlow (Zhang et al., 2025c) uses Gymnasium MuJoCo together with Franka Kitchen (Gupta et al., 2020) and Robomimic (Mandlekar et al., 2022), and DPPO (Ren et al., 2025) spans both protocols. The absence of a common benchmark makes head-to-head comparison difficult. We therefore view benchmark standardization across diffusion-RL and flow-RL methods as a near term community priority.

We also note that **Meta-World** (Yu et al., 2020), a widely used multi-task manipulation benchmark in the diffusion policy literature, has not yet been broadly adopted by flow matching RL methods, most of which evaluate on locomotion or navigation tasks instead. Extending flow matching policy evaluations to Meta-World would strengthen comparability with diffusion based baselines. Table 6 in Section 10 collects the available results for the above online RL methods. Also to be noted that the GRPO family of flow matching methods is also online but is evaluated on text-to-image generation benchmarks rather than continuous control environments and is therefore covered in the following paragraph.

**Alignment and Preference-Tuning Benchmarks.** For preference alignment and generative fine tuning, evaluation protocols differ substantially from classical control. The GRPO family of flow matching methods (detailed in Section 5.3) is primarily evaluated on text-to-image tasks using three benchmark families. **GenEval** (Ghosh et al., 2023) assesses compositional image generation by testing object co-occurrence, counting, spatial positioning, color accuracy, and attribute binding, reporting an overall accuracy score across six sub-tasks. **PickScore** (Kirstain et al., 2023) evaluates human preference alignment using a CLIP-based scoring function trained on large scale pairwise human comparisons of generated images. **Visual text rendering** measures whether text specified in the prompt appears correctly in the generated image, typically scored by OCR based edit distance following the protocol introduced in Flow-GRPO (Liu et al., 2025a). In addition, some papers report image quality and preference metrics on held-out prompt sets such as Draw-Bench (Saharia et al., 2022), including ImageReward (Xu et al., 2023a), Aesthetic Score (Schuhmann et al., 2022), and UnifiedReward (Wang et al., 2026), which serve as secondary indicators of reward hacking and generation quality degradation. Table 7 in Section 10 collects reported results across these benchmarks for the GRPO family.

**Scope of Action Spaces.** The continuous control benchmarks reviewed above (D4RL, OGBench, Mu-JoCo, DeepMind Control) all involve continuous action spaces, while the generative alignment benchmarks (GenEval, PickScore, OCR) operate in continuous latent spaces. Discrete action benchmarks such as Atari (Bellemare et al., 2013) and MinAtar (Young & Tian, 2019) are largely absent from the flow matching RL literature because flow matching is naturally formulated as a continuous density model on Euclidean spaces, and discrete flow matching variants have not yet been widely adopted as RL policy classes. Extending flow matching policies to discrete or hybrid action spaces remains an open direction.

**Evaluation Metrics Beyond Return.** Besides task performance, this survey emphasizes **inference efficiency**. For generative policies, one of the most important metric is often the *number of function evaluations* (NFE), i.e., the number of denoising or ODE-solver steps required to generate a single action. This metric is especially relevant for control because latency directly affects deployability in real-time systems. Diffusion based policies usually require iterative multi-step denoising, whereas flow based policies can often operate with substantially fewer ODE steps and, in some cases, with one-step distilled policies (Wang et al., 2023; Koirala & Fleming, 2025; Park et al., 2025b).

In addition, several recent papers report *wall-clock training time*, *wall-clock inference time*, or *step time in milliseconds*. Such measurements are valuable because two methods with similar return may differ significantly in practical deployment cost. For example, FQL explicitly reports training and inference run times under a unified implementation, showing that runtime comparisons can complement normalized return comparisons when evaluating generative policy classes (Park et al., 2025b).

A fair comparison across papers should therefore consider at least three axes simultaneously. The first is task performance such as normalized return or success rate. The second is inference cost such as NFE or milliseconds per action. The third is the benchmark family itself, since results from D4RL, OGBench, online continuous control, and preference alignment are not directly interchangeable. We follow this principle in the comparative summary in the following section.

## 10 Comparative Analysis: Flows, Diffusion, and Gaussians

To fully contextualize the rise of flow matching in reinforcement learning, we must compare it against the two dominant paradigms it seeks to replace or improve upon. These are standard unimodal policies such as Gaussian actors, and diffusion based policies. This section evaluates these families across four critical dimensions, namely representational expressivity, trajectory geometry, inference latency, and optimization stability. Table 4 further expands these dimensions into other concrete properties (e.g., path curvature and support bounding) that help clarify the practical differences between Gaussian, diffusion, and flow based policies. Beyond the conceptual distinctions discussed below, the literature also differs substantially in benchmark families and evaluation protocols. To help interpret empirical claims more carefully, Section 9 summarized the most common benchmark suites and reporting practices, and Tables 5, 6 and 7 in this section collect and report representative empirical comparisons from key diffusion and flow based RL papers.

### 10.1 Representational Expressivity: The Multimodal Imperative

Classical continuous control algorithms typically parameterize the policy as a diagonal Gaussian distribution, $\pi(a|s) = \mathcal{N}(\mu_\theta(s), \Sigma_\theta(s))$. While analytically tractable and easy to optimize via REINFORCE or reparameterization, Gaussian policies face challenges and limitations in offline RL and multi-agent settings (Lee et al., 2025a). When a dataset contains heterogeneous multimodal behaviors such as navigating around an obstacle by going either left or right, a Gaussian policy may average the modes, predicting a catastrophic "mean" action such as crashing directly into the obstacle.

Generative policies, both diffusion and flow matching solve this by learning arbitrary distributions capable of representing multiple distinct modes simultaneously. In terms of pure density estimation and expressivity, flow matching and continuous-time diffusion are theoretically equivalent, as both can universally approximate smooth target distributions. The divergence between them lies entirely in *how* they transport probability mass.

## 10.2 Trajectory Geometry: Stochastic SDEs vs. Deterministic ODEs

The most fundamental distinction between diffusion models and flow matching lies in the geometry of their sampling trajectories.

**Diffusion Models (Stochastic Transport).** Diffusion relies on a stochastic differential equation (SDE) that injects and removes Brownian noise. The path from the base distribution to the target action is inherently jagged and non-linear. While this stochasticity acts as a regularizer during image generation, it makes the sampling process in RL difficult to control and strictly binds the model to specific noise schedules (e.g., cosine or linear).

**Flow Matching (Deterministic Transport).** Flow matching learns a deterministic ordinary differential equation (ODE) represented by a continuous vector field. Crucially, variants like Rectified Flow (Liu, 2022; Liu et al., 2023) explicitly encourage *straight-line paths* between the noise $x_0$ and the target action $x_1$. This deterministic geometry provides two major advantages for decision-making:

1. **Support Constraints:** By eliminating SDE noise injection, flow models are less prone to drifting into out-of-distribution (OOD) action spaces. Methods like ReFORM (Zhang et al., 2026) exploit this by applying reflected boundary conditions directly to the ODE, mathematically guaranteeing that generated actions remain within the safe support of the training data.

2. **Action Consistency:** In planning (Zheng et al., 2023), deterministic paths ensure that slight perturbations in the latent noise do not result in wildly different action sequences, making temporal abstraction much easier.

## 10.3 Inference Latency: The NFE Bottleneck

The primary barrier to deploying generative policies in real-time robotics is inference latency, typically measured by the Number of Function Evaluations (NFE) required per environment step.[1]

- **Diffusion Policies** often require 5 to 100 NFE to generate a single action. While fast solvers (e.g., DDIM, DPM-Solver) exist, applying them in the low-NFE regime (e.g., $\leq 5$ steps) generally destroys the structural integrity of the action, leading to performance drops in RL benchmarks (Wang et al., 2023).

- **Flow Policies**, owing to their straight-line optimal transport paths, often have local truncation errors that are lower than diffusion models. Consequently, they can often be sampled accurately using fewer integration steps, corresponding to larger solver step sizes. Methods like FQL (Park et al., 2025b) and SSCP (Koirala & Fleming, 2025) demonstrate that flow policies can be distilled or evaluated in as few as 1 to 4 NFE with minimal degradation in expected return.

For high-frequency control loops ($> 50$ Hz), flow matching currently offers the only viable path for deploying un-distilled generative policies. These empirical trends are summarized quantitatively in Table 5.

## 10.4 Optimization Stability and Gradient Estimation

The final dimension of comparison is how these models interact with online reinforcement learning objectives. Updating a generative policy requires estimating the gradient of the expected return, $\nabla_\theta J(\pi_\theta)$.

In diffusion models, differentiating through an unrolled SDE sampling chain suffers from high-variance gradients, forcing researchers to rely on restrictive reward-weighted regression (Fan et al., 2025b) or complex contrastive forward-process bounds (Zheng et al., 2025b).

---

[1]The role of NFE is not symmetric across offline and online RL. In offline RL the relevant NFE is the per-action inference cost, since training does not require differentiating through the sampler. In online RL the same NFE additionally appears inside every gradient update, since methods like DPPO backpropagate through every denoising step, which is why ReinFlow's noise-injection trick at one denoising step matters and why DPPO's ten denoising steps incur a real wall-clock cost during training. Tables 5 and 6 should be read with this distinction in mind.

Because flow matching relies on deterministic ODEs, it provides cleaner pathways for gradient estimation. As demonstrated by PolicyFlow (Yang et al., 2026), the importance ratio between an old and updated CNF policy can be approximated using a Gaussian likelihood-ratio identity together with a velocity-field interpolation along a simple linear path, avoiding the need to fully simulate the ODE during training. This enables flow models to be optimized using standard clipped objectives such as Proximal Policy Optimization (PPO), bridging the gap between highly expressive policy parameterizations and the reliability of classical actor-critic algorithms.

## 10.5 Summary of Trade-offs and Empirical Comparisons

Table 4 summarizes these architectural trade-offs. While Gaussian policies remain the most computationally efficient, Flow Matching offers a more favorable trade-off than Diffusion in RL contexts by offering the same multi-modal expressivity but with superior inference latency, deterministic safety bounds, and stable gradient pathways.

| Property | Gaussian Policy | Diffusion Policy | Flow-Matching Policy |
|---|---|---|---|
| **Expressivity** | Unimodal | Highly Multimodal | Highly Multimodal |
| **Transport Geometry** | N/A | Stochastic (SDE) | Deterministic (ODE) |
| **Path Curvature** | N/A | Highly Curved | Straight (Rectified) |
| **Inference Latency** | 1 NFE | 5–100 NFE | 1–10 NFE (1 if distilled) |
| **Online Optimization** | PPO / SAC (Stable) | High Variance | PPO / GRPO (Stable) |
| **Support Bounding** | Clipping | Difficult (Drift) | Exact (e.g., ReFORM) |

Table 4: Comparative summary of policy classes in continuous reinforcement learning.

**Empirical Evidence Across Benchmarks.** While the above discussion provides a conceptual comparison, recent work has also empirically quantified the trade-offs between diffusion and flow-based policies on standard offline RL benchmarks such as D4RL. In particular, prior studies consistently report that diffusion policies require higher numbers of function evaluations (NFE) during inference, whereas flow based policies can achieve comparable or superior performance with fewer steps. To make this comparison concrete, we summarize representative results from key works in Tables 5 and 6. The gap in NFE is particularly critical in real-time control, where action inference must occur at frequencies exceeding 10–50 Hz. In such regimes, diffusion-based policies often require aggressive distillation or approximation (Lu et al., 2022; Song et al., 2020), whereas flow-based policies can be deployed directly with minimal numerical integration overhead. Notably, a well tuned Gaussian baseline like ReBRAC (Tarasov et al., 2023) remains competitive with the best diffusion and flow methods on D4RL Gym-Loco, suggesting that the practical advantage of expressive generative policies is most visible on benchmarks with highly multimodal action distributions such as AntMaze, OGBench manipulation, and visual control, rather than on relatively unimodal locomotion tasks.

**Reading Table 5.** Three trends emerge from the consolidated offline numbers. First, on D4RL Gym-Loco, a well-tuned Gaussian baseline like ReBRAC (Tarasov et al., 2023) reaches an average of 88.5, which is statistically indistinguishable from Diffusion-QL (88.0) and SSCP/SSCQL (87.9), and slightly above QIPO-OT (86.3). This suggests that the practical gain from expressive generative policies is modest on relatively unimodal locomotion data, and is most visible on benchmarks with strongly multimodal action distributions such as AntMaze, OGBench manipulation, and visual control. Second, the AntMaze column tells a different story. FQL reaches 84 with a single-step actor, IDQL reaches 79.1 but uses 128 rejection samples per action, and ReFORM reaches 81 with a reflected ODE that enforces support constraints by construction. The latency gap between these methods is substantial in the deployment regime, and a one-step distilled flow policy like FQL becomes the most attractive option once one accounts for action-generation cost. Third, energy-weighted flow methods such as QIPO-OT and QIPO-Diff explicitly report per-action wall-clock times of 27 to 56 milliseconds, compared to 75 milliseconds for QGPO, providing the kind of efficiency evidence the survey emphasizes throughout. The OGBench column shows the same gap more dramatically. FQL

Table 5: **Consolidated empirical comparison of diffusion-based and flow-matching offline RL methods.** NFE = number of velocity-field / denoiser evaluations per action at inference. D4RL Gym-Loco is the 9-task halfcheetah/hopper/walker2d {medium, medium-replay, medium-expert} v2 average, D4RL AntMaze is the 6-task umaze/medium/large {play, diverse} average. The OGBench column reports the 50-task state-based average across 5 locomotion and 5 manipulation environments (5 tasks each), with all baselines re-implemented in a single codebase by Park et al. (2025b) Table 2. Numbers are normalized scores (%) as reported by each paper under its own protocol; this is not a head-to-head reproduction. "–" indicates the paper does not report on that benchmark, "↓" for time means lower is better. For methods that use rejection sampling (IDQL, IFQL), NFE is reported as $T \times N$, where $T$ is the number of denoising or ODE steps per candidate action and $N$ is the number of candidates scored by the Q-function.

| Method | Class | NFE | D4RL (normalized return, %) | | OGBench | Action gen. | Notes / source |
|---|---|---|---|---|---|---|---|
| | | | Gym-Loco (avg) | AntMaze (avg) | (50-task avg) | time (↓) | |
| *Gaussian / non-generative baselines (for context)* | | | | | | | |
| BC | Gaussian | 1 | 51.9 | 16.7 | 2.9 | – | Wang et al. (2023) |
| TD3+BC | Gaussian | 1 | 75.3 | 27.3 | – | – | Wang et al. (2023) |
| IQL | Gaussian + AWR | 1 | 77.0 | 63.0 | 23.4 | – | 20 min train (D4RL) (Hansen-Estruch et al., 2023) |
| **ReBRAC** | Gaussian | 1 | 88.5[*] | 76.8 | 31.0 | – | [*]9-task v2 avg. from Tarasov et al. (2023) Tab. 16 |
| *Diffusion-based offline RL* | | | | | | | |
| Diffuser | Diff. trajectory | 100 | 75.3 | – | – | – | 100 denoising steps (Janner et al., 2022; Wang et al., 2023) |
| Diffusion-QL | Diff. policy + Q | 5 | 88.0 | 69.6 | – | – | N=5 best on D4RL (Wang et al., 2023) |
| IDQL | Diff. + rejection | $5 \times 128$ | 82.1 | 79.1 | 23.0 | 60 min train | N=128 samples, ∼4× faster than DQL (Hansen-Estruch et al., 2023) |
| QGPO | Diff. + energy guided | – | – | 78.3 | – | 75.05 ms | Per-action time (Zhang et al., 2025b) |
| QIPO-Diff | Diff. + Q-weighting | low-step | 86.2 | 77.3 | – | 55.86 ms | −26% per-action time vs. QGPO (Zhang et al., 2025b) |
| *Flow-matching offline RL* | | | | | | | |
| QIPO-OT (EFM) | FM + Q-weighting | low-step | 86.3 | 72.0 | – | 27.26 ms | −64% per-action time vs. QGPO (Zhang et al., 2025b) |
| SSCP (SSCQL) | 1-step FM | 1 | 87.9 | – | – | – | Single-step generation (Koirala & Fleming, 2025) |
| FBRAC | Flow + BPTT | 10 | – | 64[†] | 28.6 | high (BPTT) | [†]D4RL antmaze (Park et al., 2025b) |
| IFQL | Flow + rejection | $10 \times 32$ | – | 65[†] | 30.2 | high (rej. samp.) | [†]D4RL antmaze (Park et al., 2025b) |
| FAWAC | Flow + AWR | 10 | – | 44[†] | 16.0 | moderate | [†]D4RL antmaze (Park et al., 2025b) |
| **FQL** | Flow prior + 1-step actor | **1** | – | **84**[†] | **43.8** | lowest among flow-based | [†]D4RL antmaze (Park et al., 2025b) |
| ReFORM | Reflected FM | 10 | – | 81[‡] | – | moderate | [‡]ReFORM Tab. 5, ∼80 min/$10^6$ steps (Zhang et al., 2026) |

achieves 43.8 across 50 state-based tasks, compared to 31.0 for the strongest Gaussian baseline (ReBRAC) and 23.0 to 30.2 for the diffusion and other flow baselines, with a single function evaluation per action versus 10 to 640 for rejection-sampling alternatives. This widens the practical advantage of one-step distilled flow policies on benchmarks specifically designed to expose the limits of unimodal policy classes.

Table 6: **Online RL with flow-matching and diffusion policies.** The two blocks correspond to the two evaluation protocols used in the literature. **(Top)** MuJoCo Playground / DeepMind Control Suite, training from scratch with 60M environment steps where values are mean evaluation reward ± standard error averaged across 10 DMC tasks (5 seeds each), reported in McAllister et al. (2025) Table 1. **(Bottom)** OpenAI Gym MuJoCo, offline-pretrained then online fine-tuned (∼$8 \times 10^7$ env. samples) where values are mean episode reward ± std. across 3 seeds, reported in Zhang et al. (2025c) Table 4(a). Pre-trained → fine-tuned shows the gain from online RL on a flow-matching policy. PolicyFlow numbers are described qualitatively because Yang et al. (2026) report results as learning curves (Figure 3) without a numeric aggregate table and their text explicitly states PolicyFlow "achieves performance comparable to or exceeding FPO in most environments, outperforming DPPO".

| Method | Class | NFE | Reported reward | | | | Notes / source |
|---|---|---|---|---|---|---|---|
| *MuJoCo Playground / DMC, 10-task average eval reward (60M env. steps)* | | | | | | | |
| Gaussian PPO | Gaussian (PPO) | 1 | 667.8 ± 66.0 | | | | Tuned Brax PPO baseline (McAllister et al., 2025) |
| DPPO | Diffusion + PPO | 10 | 652.5 ± 83.7 | | | | 10 denoising steps, $\sigma_t = 0.05$ (McAllister et al., 2025) |
| **FPO** | Flow + advantage-weighted PG | 10 | **759.3 ± 45.3** | | | | 8 $(\tau, \epsilon)$ pairs, $\epsilon$-MSE (McAllister et al., 2025) |
| PolicyFlow | CNF + PPO + Brownian reg. | 10–12 | matches/exceeds FPO, > DPPO > PPO | | | | 8 DMC tasks at 30M steps (Yang et al., 2026) |
| *OpenAI Gym, pre-trained → fine-tuned episode reward (BC pretraining → online RL)* | | | | | | | |
| | | | Hopper-v2 | Walker2d-v2 | Ant-v2 | Humanoid-v3 | |
| ReinFlow-R | Rectified Flow + PPO + noise inj. | 1 | 1432 → 3205 | 2740 → 4109 | 1231 → 4009 | 1926 → 5076 | 1 denoising step (Zhang et al., 2025c) |
| ReinFlow-S | Shortcut FM + PPO + noise inj. | 1–4 | 1528 → 3283 | 2739 → 4255 | 2088 → 4106 | 2122 → 4749 | 1–4 steps, best wall-time (Zhang et al., 2025c) |

**Reading Table 6.** The online block highlights a methodological caveat. The top half compares methods that train from scratch on MuJoCo Playground, where FPO outperforms both Gaussian PPO and DPPO on 8 of 10 DM Control tasks and PolicyFlow further matches or exceeds FPO on a subset of 8 tasks. The

bottom half reports a different protocol entirely, namely behavior-cloning pretraining followed by online fine-tuning on Gymnasium MuJoCo, where ReinFlow lifts pretrained rewards by between 50 and 226 percent depending on task difficulty. Because the two settings differ in initial policy quality, training budget, and benchmark family, a direct numerical ranking across all four methods would be misleading. What the table does show clearly is that flow matching policies under PPO-style optimization are now competitive with or stronger than diffusion based and Gaussian baselines on every continuous control suite where they have been evaluated, and that fine tuning a flow-matching policy with online RL recovers most of the headroom left by behavior cloning even at one denoising step.

| Method | Steps | GenEval ↑ | OCR Acc. ↑ | PickScore ↑ | Focus |
|---|---|---|---|---|---|
| SD3.5-M (base) | — | 0.63 | 0.59 | 21.72 | — |
| Flow-GRPO (Liu et al., 2025a) | 5,600 | 0.95 | 0.92 | 23.31 | Baseline |
| TempFlow-GRPO (He et al., 2025) | 3,800 | **0.97** | — | — | Temporal credit |
| AWM (Xue et al., 2025) | —[†] | 0.95 | 0.95[†] | 23.25[†] | 8–24× speedup |
| SuperFlow (Chen et al., 2025c) | 1,846[‡] | 0.80[‡] | 0.84[‡] | 0.87[*] | Efficiency |
| Smart-GRPO (Yu et al., 2026) | 360[§] | — | — | —[**] | Noise optimization |
| GRPO-Guard (Wang et al., 2025b) | 1,860[‡] | 0.95[‡] | 0.93[‡] | — | Over-optimization |

Table 7: **Reported results for the GRPO family on text-to-image alignment (SD3.5-M base).** All numbers are taken from the respective original papers and are not reproduced under identical evaluation conditions. Empty cells indicate metrics not reported by that method on this benchmark. [†]AWM reports GPU hours rather than steps and reaches the listed scores with extended training (79 and 205 GPU hours for OCR and PickScore respectively). [‡]SuperFlow and GRPO-Guard evaluate at step counts matched to Flow-GRPO to demonstrate efficiency or stability gains, not converged final performance. [*]SuperFlow reports PickScore on a different normalization scale and is not directly comparable to the raw PickScore values in other rows. [**]Smart-GRPO reports ImageReward (0.86) and Aesthetic Score (6.24) instead of the three benchmarks above, evaluated on a 1,000-prompt GenEval-script set. [§]Smart-GRPO reports in epochs rather than steps.

**Reading Table 7.** All methods in this table use Stable Diffusion 3.5-Medium as the base model and are evaluated on three standard text-to-image benchmarks: GenEval (Ghosh et al., 2023) for compositional image generation, OCR accuracy for visual text rendering (Liu et al., 2025a), and PickScore (Kirstain et al., 2023) for human preference alignment. These numbers are collected from individual papers and are not head-to-head reproductions under identical conditions, so cross-row comparisons should be treated as indicative rather than definitive. Among methods that target final task performance, TempFlow-GRPO achieves the highest reported GenEval score (0.97 in 3,800 steps versus 0.95 for Flow-GRPO at 5,600 steps), and AWM matches Flow-GRPO on all three benchmarks while reporting 8–24× wall-clock speedups, reaching 0.95 OCR accuracy and 23.25 PickScore with extended training. SuperFlow and GRPO-Guard evaluate at step counts matched to Flow-GRPO to demonstrate efficiency and stability improvements respectively, so their numbers reflect advantages under same compute conditions rather than converged final performance. Smart-GRPO reports on ImageReward (Xu et al., 2023a), and Aesthetic Score (Schuhmann et al., 2022) rather than the three benchmarks above, achieving the highest values on both metrics among the GRPO methods it compares against.

## 11 Open Problems and Future Directions

While flow matching has demonstrated clear advantages over diffusion models in terms of inference speed and deterministic stability, its integration with reinforcement learning is still in its infancy. In this section, we outline the most pressing theoretical and practical challenges that represent fertile ground for future research.

## 11.1 Principled Exploration in Deterministic Flows

The most fundamental friction between RL and flow matching lies in exploration. Standard RL relies on stochastic policies (e.g., Gaussian noise) to discover high-reward regions. However, flow matching models learn deterministic ODEs. Current online methods address this by injecting arbitrary Gaussian noise into the ODE solver (Zhang et al., 2025c) or converting the ODE to an SDE (Liu et al., 2025a). While these heuristics work, they destroy the straight-line optimal transport properties that make flow matching efficient in the first place.

**Open Problem:** How can we design *exploration-aware vector fields*? Future work can investigate methods that structurally embed exploration into the flow itself; for example, by learning a distribution over vector fields (Bayesian Flow Networks) or utilizing optimistic initialization of the base distribution $p_0$, rather than relying on post-hoc SDE conversions that introduce noise artifacts.

## 11.2 Off-Policy and Continuous-Time RL Integration

Most successful online flow-matching RL algorithms, such as PolicyFlow (Yang et al., 2026) and FPO (McAllister et al., 2025), are built on on-policy algorithms like PPO. This is because evaluating the exact likelihood of a flow-generated action is computationally heavy, making off-policy methods (which require evaluating old actions under the current policy) prohibitively slow.

**Open Problem:** Scaling flow matching to sample-efficient, off-policy continuous control (e.g., Soft Actor-Critic equivalents for flows) remains unsolved. Furthermore, because ODEs operate in continuous time ($t \in [0,1]$ for the flow), there is an unexplored theoretical opportunity to bridge generative flow steps with continuous-time MDPs, treating the generation process itself as a sequential decision-making problem governed by Bellman equations.

## 11.3 Scaling to High-Dimensional Visuomotor Control

Currently, many flow-RL benchmarks focus on state-based control or relatively simple image-based tasks. However, the ultimate goal of generative policies is to scale Vision-Language-Action (VLA) models for generalist robotics. While methods like ARFM (Zhang et al., 2025a) begin to address this, flow matching models often struggle with the "curse of dimensionality" when the state/context vector (high-res images) is vastly larger than the action vector.

**Open Problem:** How can we build efficient conditional flow architectures for pixel-to-action tasks? Research is needed into latent flow matching for RL, where the RL objective operates in a compressed latent space rather than the raw action space and the development of Transformer-based architectures (like DiT) specifically optimized for the high-frequency control loops required by real-world robotics.

## 11.4 Hardware-Aware Solvers and Single-Step Distillation

Even though flow matching requires significantly fewer Number of Function Evaluations (NFE) than diffusion, running an ODE solver for 3 to 5 steps per environment interaction is still too slow for many real-time robotic systems operating at high frequencies.

**Open Problem:** While algorithmic distillation techniques like SSCP (Koirala & Fleming, 2025) offer single-step generation, they may be challenged by capacity bottlenecks, losing the multimodal expressivity of the full ODE. Future directions can explore hardware-aware neural ODE solvers and progressive distillation techniques that can dynamically trade off between inference latency and action precision depending on the urgency of the robot's current state.

## 11.5 Alignment Beyond Scalar Rewards

As flow matching replaces diffusion in alignment tasks (RLHF) for foundation models (Zhu et al., 2025; Sun et al., 2025), the reliance on scalar reward models becomes a bottleneck. Scalar rewards often lead to mode-collapse, where the model only generates the single "safest" response.

**Open Problem:** Can flow matching directly optimize for multi-dimensional preferences or human-in-the-loop feedback without intermediate scalar reward models? Techniques like Preference Flow Matching (PFM) (Kim et al., 2024) hint at this by treating alignment as a transport problem from "rejected" to "preferred" distributions, but scaling this geometrically-grounded alignment to diverse reasoning and coding tasks remains an open frontier.

## 12 Conclusion

The integration of generative models into reinforcement learning has fundamentally expanded the capabilities of autonomous agents, allowing them to model highly complex, multimodal behaviors. As this survey demonstrates, the field is currently undergoing a critical transition from diffusion-based models to *flow matching* architectures. By replacing stochastic denoising processes with deterministic, optimal transport guided ODEs, flow matching resolves the primary bottlenecks of generative RL: it reduces inference latency, provides structurally bounded generation paths, and enables stable gradient estimation for online policy optimization.

We provided a unified taxonomy to organize this rapidly expanding literature, categorizing methods along two orthogonal axes: the distribution being modeled (ranging from single-step actions to transition dynamics and critic values) and the mechanism of RL signal integration (from offline energy-weighted regression to online group relative policy optimization). This framework clarifies the deep theoretical connections between seemingly disparate algorithms, illustrating how deterministic probability transport serves as a general-purpose tool for decision-making.

While challenges remain, particularly concerning principled exploration, off-policy scaling, and hardware-aware acceleration, the trajectory of the research is clear. Flow matching may become the standard generative backbone not only for continuous control and robotics but also for the alignment of large-scale foundation models. We hope this survey serves as a conceptual foundation and a practical roadmap for researchers interested in flow matching in reinforcement learning.

### Broader Impact Statement

This survey systematizes recent algorithmic advances in flow matching for reinforcement learning. While the foundational nature of this work does not directly introduce new societal risks, the algorithms discussed herein are actively used to train autonomous systems, robotics, and large language models. Improvements in the sample efficiency and alignment of these models carry dual-use implications. Ensuring that reward-guided flow models are aligned with safe and ethical human preferences, particularly as these models are deployed in physical environments or user-facing AI systems remains a critical responsibility for the research community.

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

# A  Diffusion-Based Reinforcement Learning

## A.1  Overview of Diffusion in RL methods

Most diffusion-RL methods can be organized by **what object is generated** and **how RL objectives enter**:

**(1) Diffusion as a planner (trajectory / action-sequence generation).**  Here, the model generates an entire sequence of future states and actions $\tau = (s_t, a_t, \ldots, s_{t+H}, a_{t+H})$ simultaneously, treating planning as a data imputation problem. A canonical instance is Janner et al. (2022) introducing *Diffuser*, which uses a temporal U-Net to denoise full trajectories, guiding the process with separate reward classifiers. Subsequent works study planner design choices and extensions such as hierarchical diffusion planning, latent-space planning, trajectory stitching, and prior/guidance mechanisms (Chen et al., 2024; Li, 2024; Lu et al., 2025; Zeng et al., 2025; Ki et al., 2025; Lee & Choi, 2025; Ni et al., 2023). Diffusion planners have also been explored in domains like autonomous driving and traffic simulation (Zheng et al., 2025c; Zhong et al., 2023).

**(2) Diffusion as a policy (one-step action generation).**  These methods use the diffusion model strictly as a conditional policy $\pi(a|s)$, generating a single action $a_t$ given the current state $s_t$. In offline RL, *Diffusion-QL* (Wang et al., 2023) and *IDQL* (Hansen-Estruch et al., 2023) combine diffusion actors with Q-learning critics to handle multi-modal datasets without the extrapolation error of standard behavior cloning, with many variants targeting robustness and generalization (Ada et al., 2024; Yang & Xu, 2024; Liu et al., 2025b; Sun & Zheng, 2024). In robotics, action diffusion has become a popular policy class for visuomotor control and multi-task manipulation, including extensions to 3D and hierarchical policies (Chi et al., 2023; He et al., 2023a; Ze et al., 2024; Ma et al., 2024).

**(3) Diffusion for constrained/safe RL (feasibility and constraints).**  Several papers incorporate safety/feasibility constraints into the generative process, either as explicit guidance signals or via constrained policy optimization formulations (Fang et al., 2025; Zheng et al., 2024; He et al., 2023b).

**(4) Diffusion for imitation learning, skills, and human-in-the-loop control.**  Beyond offline RL benchmarks, diffusion models are used for goal-conditioned imitation and skill learning, shared autonomy, and human-robot collaboration (Reuss et al., 2023; Jain & Ravanbakhsh, 2024; Yoneda et al., 2023; Ng et al., 2023; Chen et al., 2023a; Xu et al., 2023b). Data augmentation and behavior retargeting via generative models is another recurring theme (Chen et al., 2023c; Lu et al., 2023b; Yu et al., 2023).

**(5) Diffusion for world models and environment simulation.**  Diffusion can be used as a dynamics/world model that generates plausible futures, especially in partially observed or visually rich settings. Representative examples include diffusion world modeling in Atari and improved memory/imagination consistency in diffusion world models (Alonso et al., 2024; Lee et al., 2025b; Rigter et al., 2024; Zhang et al., 2024).

**(6) Diffusion + preference/reward modeling and RL fine-tuning.**  A parallel line uses RL-style training to align diffusion models with preferences or task rewards. Examples include preference/customization for diffusion models and reward-driven training pipelines (Dong et al., 2024; Black et al., 2024; Liu et al., 2024; Yoon et al., 2024; Zhang et al., 2023). (These works are conceptually close to RLHF-style alignment, but appear in diffusion-RL bibliographies because they explicitly optimize diffusion generation using RL signals.)

### A.2   How RL objectives enter diffusion models: training-time vs sampling-time

A second critical distinction is **when** the RL objective (reward/value signal) is applied:

**Training-time integration.**   Some methods incorporate RL signals directly into learning (e.g., value-weighted objectives, constrained updates, or actor-critic training with diffusion as the policy class), including diffusion actor-critic variants and diffusion policy-gradient style training (Wang et al., 2024; Li et al., 2024; Fang et al., 2025).

**Sampling-time guidance.**   Other methods keep the base diffusion model fixed and bias denoising using an external score/energy/value signal. Energy-guided diffusion for offline RL is a key example (Lu et al., 2023a). Related guidance ideas appear in planning settings (e.g., prior-guided planning and flexible guidance in driving) (Ki et al., 2025; Zheng et al., 2025c).

### A.3   Efficiency and fast sampling

A core deployment issue is that diffusion control is often *too slow* if many denoising steps are needed per environment step. Accordingly, diffusion-RL papers frequently adopt faster samplers/solvers and/or distill the reverse process. Learning-free accelerations include DDIM (Song et al., 2020) and high-order ODE solvers such as DPM-Solver (Lu et al., 2022). Learning-based acceleration includes shortcut/few-step training and distillation methods such as consistency-style objectives and policy classes (Song et al., 2023; Ding & Jin, 2024; Chen et al., 2023b; Fan & Lee, 2023). For offline RL specifically, efficiency-driven diffusion policy variants reduce backpropagation through long chains or approximate sampling during training (Kang et al., 2023; Venkatraman et al., 2024).

### A.4   From Diffusion to Flow Matching

Despite their expressivity, diffusion models face deployment challenges in control:

1. **Inference Latency:** Standard diffusion policies require 10–100 denoising steps to generate a single action. While fast solvers like DDIM (Song et al., 2020) or DPM-Solver (Lu et al., 2022) reduce this, they often degrade sample quality in low-step regimes (e.g., $< 5$ steps).

2. **Stochasticity vs. Control:** The stochastic nature of the reverse process (SDE) complicates the use of precise solvers and makes it difficult to guarantee smooth control updates.

These limitations specifically the conflict between high-quality generation and real-time inference speed directly motivate the adoption of **Flow Matching in RL**, which we detail in this survey. Flow Matching offers the expressivity of diffusion but with deterministic, straight-line trajectories that enable faster and more stable policy learning.

