# OpenReview forum: "A Survey of Flow Matching in Reinforcement Learning"
_TMLR — Accepted by TMLR_

### Review · Reviewer_2L9P · 2026-03-30

**Summary Of Contributions:**

This paper provides a timely and well-structured survey on the integration of Flow Matching (FM) techniques within Reinforcement Learning (RL). Rather than list existing literature, the authors contribute a thoughtful taxonomy that dissects the field along two critical axes: the specific RL components modeled by FM (e.g., policy representation, transition dynamics) and the mechanism of RL signal integration (e.g., energy-weighted regression, flow-based policy gradients).

The primary contribution lies in conceptually positioning FM not just as a special case of diffusion models, but as a structural framework for RL due to its deterministic ODEs. By highlighting how FM addresses the latency and integration bottlenecks of SDEs, the paper offers a unifying perspective that bridges modern generative models with RL.

**Audience:**

Yes

**Audience Explanation:**

The intersection of generative AI and Reinforcement Learning is currently one of the most active and rapidly evolving areas in machine learning. While diffusion models have recently dominated generative RL, their inference latency remains a well-known pain point for real-time control. A comprehensive survey that organizes this field, clarifies the mathematical transition from diffusion to FM in RL, and list open problems will serve as a highly valuable reference.

**Claims And Evidence:**

Yes

**Claims Explanation:**

For a survey paper, "evidence" is primarily derived from the comprehensiveness of the literature review, the logical soundness of the proposed classification, and the theoretical grounding of comparative analyses. The authors successfully support their claims regarding FM's computational and representational advantages by dissecting the underlying mathematics and explicitly linking these mathematical properties to RL challenges. The comparative analysis against Gaussian and Diffusion-based baselines is technically sound and well-reasoned, relying on established findings in the cited literature to substantiate the claim that FM serves as a highly efficient generative modeling framework for diverse behavioral strategies.

**Requested Changes:**

Please include a consolidated summary table that compares the reported empirical results of key FM-RL and Diffusion-RL papers. Highlighting metrics such as inference latency (e.g., Number of Function Evaluations, NFE) alongside general performance on standard benchmarks will make the efficiency claims much more concrete.

In addition, I would also recommend that the author include an introduction to benchmark datasets and evaluation methods in related fields, as this is essential for readers to quickly conduct relevant experiments.

---

> ### Author Response · Authors · 2026-05-11
>
> We sincerely thank the reviewer for the helpful feedback, and especially for the suggestions to include a consolidated empirical comparison and a discussion of benchmarks and evaluation protocols. We have addressed both points in the revised manuscript.
>
> > “Please include a consolidated summary table that compares the reported empirical results of key FM-RL and Diffusion-RL papers. Highlighting metrics such as inference latency (e.g., Number of Function Evaluations, NFE) alongside general performance on standard benchmarks will make the efficiency claims much more concrete.”
>
> We have added three consolidated empirical tables in the revised Section 10 (Comparative Analysis).
>
> **Table 5** provides a consolidated comparison of diffusion-based and flow-matching offline RL methods. It reports method class, NFE per action at inference, normalized return on D4RL Gym-Loco (the 9-task halfcheetah/hopper/walker2d medium/medium-replay/medium-expert v2 average), D4RL AntMaze (the 6-task umaze/medium/large play/diverse average), and OGBench (the 50-task state-based average), along with per-action generation time in milliseconds. The table includes Gaussian baselines (BC, TD3+BC, IQL, ReBRAC) for context, diffusion methods (Diffuser, Diffusion-QL, IDQL, QGPO, QIPO-Diff), and flow-matching methods (QIPO-OT/EFM, SSCP/SSCQL, FBRAC, IFQL, FAWAC, FQL, ReFORM). For methods using rejection sampling (IDQL, IFQL), NFE is reported as $T \times N$ where $T$ is the number of steps per candidate and $N$ is the number of candidates scored by the Q-function.
>
> We include a "Reading Table 5" paragraph synthesizing three key trends. First, on Gym-Loco, ReBRAC (88.5) is statistically indistinguishable from Diffusion-QL (88.0) and SSCP (87.9), suggesting expressive generative policies are most advantageous on benchmarks with strongly multimodal action distributions such as AntMaze and OGBench. Second, on AntMaze, FQL reaches 84 with a single-step actor while IDQL reaches 79.1 but uses 128 rejection samples per action. The latency gap is substantial in deployment, making one-step distilled flow policies particularly attractive. Third, on OGBench, FQL achieves 43.8 compared to 31.0 for ReBRAC and 23.0 to 30.2 for diffusion and other flow baselines, widening the practical advantage on benchmarks designed to expose the limits of unimodal policies. QIPO-OT and QIPO-Diff also report per-action wall-clock times of 27 to 56 milliseconds compared to 75 milliseconds for QGPO, providing concrete efficiency evidence.
>
> **Table 6** covers online RL in two blocks. The top block compares methods training from scratch on MuJoCo Playground and DMC (Gaussian PPO, DPPO, FPO/FMPG, PolicyFlow) with 60M steps. The bottom block covers offline-pretrained then online-fine-tuned methods on Gym MuJoCo (ReinFlow). The "Reading Table 6" paragraph notes that direct ranking across protocols would be misleading, but flow-matching policies under PPO-style optimization are competitive with or stronger than both baselines on every suite evaluated.
>
> **Table 7** reports GRPO family results on text-to-image alignment using SD 3.5-Medium (GenEval, OCR, PickScore). TempFlow-GRPO achieves the highest GenEval score (0.97 in 3,800 steps versus 0.95 for Flow-GRPO at 5,600 steps), and AWM matches Flow-GRPO with 8 to 24x speedups. All captions note these are not head-to-head reproductions.
>
> > “I would also recommend that the author include an introduction to benchmark datasets and evaluation methods in related fields, as this is essential for readers to quickly conduct relevant experiments.”
>
> We have added a new Section 9 ("Benchmarks and Evaluation Protocols") covering four areas.
>
> **Offline RL Benchmarks.** We describe D4RL (Fu et al., 2020) with its Gym locomotion, AntMaze, and Adroit domains, and OGBench (Park et al., 2025a) for state-based and pixel-based tasks, noting which flow-matching papers evaluate on which suite.
>
> **Online RL and Continuous-Control Benchmarks.** We cover Gymnasium MuJoCo, MuJoCo Playground, and the DeepMind Control Suite, highlighting that flow-matching papers have not converged on a shared suite and that benchmark standardization is a near-term priority. We also note that Meta-World has not yet been broadly adopted by flow-matching RL methods.
>
> **Alignment and Preference-Tuning Benchmarks.** We describe GenEval, PickScore, and OCR-based visual text rendering, plus secondary metrics (ImageReward, Aesthetic Score, UnifiedReward).
>
> **Evaluation Metrics Beyond Return.** We discuss NFE as the key efficiency metric and wall-clock time measurements, concluding with a reporting caveat that fair comparisons require considering task performance, inference cost, and benchmark family simultaneously.
>
> We hope these additions give readers the practical orientation the reviewer requested.

---

### Review · Reviewer_XPfe · 2026-04-28

**Summary Of Contributions:**

This paper provides a very comprehensive summary of the popular methodologies to train a flow matching policy with reinforcement learning. The authors did a good job in outlining the taxonomy of this subfield, demonstrating the diversity of related research, spanning offline RL, online RL, critic and alignment, and research frontier. The weakness mainly lies in lack of including many applications, for example in robotics.

**Audience:**

Yes

**Audience Explanation:**

People that are doing training on flow models with RL, especially those in robotics and biotech and image generation.

**Broader Impact Concerns:**

This paper does not contain explicit ethical concerns.

**Claims And Evidence:**

Yes

**Claims Explanation:**

The authors introduced ReinFlow, FlowGRPO and FQL and their variants in page 12, 18, etc. They provided a concise and accurate characterization of these methods.
The paper focus more on introducing the algorithm methodologies but lacks more narration on how these methods can be applied to relevant fields.

**Requested Changes:**

Add more descipriont on the application of these methods.

---

> ### Author Response · Authors · 2026-05-11
>
> We thank the reviewer for the positive evaluation and for highlighting the need for stronger coverage of applications. We have addressed this point with a new dedicated section in the revised manuscript.
>
> > “Add more description on the application of these methods.”
>
> We have added a new Section 8 ("Applications of Flow Matching in RL"), placed after the Emerging Frontiers section (Section 7) and before the new Benchmarks and Evaluation Protocols section (Section 9). While the preceding sections of the survey are organized around algorithmic principles, Section 8 synthesizes how the methods discussed earlier are being deployed across distinct application domains, with separate paragraphs covering the following areas.
>
> **Robotics and visuomotor control.** This is the largest application area for flow matching policies. We discuss how diffusion based visuomotor policies introduced by Chi et al. (2023) and Ze et al. (2024) established expressive policy representations for real robot manipulation, and how flow matching successors target the inference latency bottleneck. Specifically, we cover SSCP for single step generation (Koirala and Fleming, 2025), FQL for one step distilled actors with flow priors (Park et al., 2025b), Riemannian Flow Matching Policies (RFMP/SRFMP) for non-Euclidean end-effector states on SE(3) (Braun et al., 2024 and Ding et al., 2025), FPO for fine tuning Vision-Language-Action models such as $\pi_0$ (Lyu et al., 2025), ARFM for addressing signal/variance tradeoffs in VLA-scale post-training (Zhang et al., 2025a), and residual policies such as EXPO (Dong et al., 2025) and RFS (Su et al., 2026) support local refinement on dexterous manipulation without disturbing the multimodal flow prior.
>
> **Image and video generation alignment.** Flow matching has rapidly become the backbone of choice for RL-based fine tuning of large generative models. The GRPO family (Flow-GRPO, TempFlow-GRPO, Smart-GRPO, SuperFlow) is primarily evaluated on text-to-image alignment with human preference reward models such as PickScore and ImageReward. GRPO-Guard, AWM, and Fine-Grained GRPO target stability and credit assignment for the same setting. DiffusionNFT applies flow matching to online RL on diffusion backbones for image generation.
>
> **Speech and audio.** F5R-TTS applies GRPO to a flow matching text-to-speech model, optimizing word error rate and speaker similarity rewards for zero-shot voice cloning (Sun et al., 2025). YingMusic-SVC extends Flow-GRPO with singing specific inductive biases for zero-shot singing voice conversion (Chen et al., 2025b).
>
> **Language model reasoning.** FlowRL reframes RL fine tuning of LLM reasoning chains as matching a target distribution of high reward trajectories rather than maximizing a scalar reward, preserving solution diversity in mathematical reasoning benchmarks (Zhu et al., 2025).
>
> **Autonomous driving and traffic simulation.** Although most diffusion based driving and traffic simulation systems have not yet been re-implemented in pure flow matching form, they share the planning as generation framing that flow matching trajectory methods such as GTP (Feng et al., 2025b) and CtrlFlow (Wang et al., 2025a) build on. We note that this framing may transfer naturally to deterministic flow based planners.
>
> **Multi-agent coordination and goal-conditioned navigation.** MAC-Flow (Lee et al., 2025a) models centralized joint action distributions and distills them into fast decentralized one step policies, demonstrating multi-agent coordination beyond standard continuous control. Goal-conditioned trajectory generation (Diao et al., 2025), Extremum Flow Matching (Rouxel et al., 2025), and Intention-Conditioned Flow Occupancy Models (Zheng et al., 2025a) apply flow matching to humanoid goal reaching, robotic navigation, and future state visitation prediction respectively.
>
> **Transfer learning under dynamics shift.** CompFlow uses flow matching as a distributional discrepancy estimator between source and target environment dynamics, leveraging the optimal transport interpretation of flow matching to guide both data selection and exploration (Kong et al., 2025).
>
> Several applications mentioned briefly in the body (for example RFMP/SRFMP, F5R-TTS, FPO) are now consolidated and cross-referenced in Section 8, making it easier for readers in robotics, biotech, and image generation to locate relevant methods. In addition, we have added a dedicated Section 9 (Benchmarks and Evaluation Protocols) that documents benchmark families and evaluation metrics across these domains, and three new empirical comparison tables (Tables 5, 6, and 7 in Section 10) that make the performance and efficiency claims concrete.
>
> We sincerely thank the reviewer again for the helpful feedback.

---

> > ### Comment · Reviewer_XPfe · 2026-05-27
> > **Reply to author's response**
> >
> > I appreciate the author's revision that mentions more applications of this subfield, which I think will broaden the audience of this paper. I think this work is ready to be published now.

---

### Review · Reviewer_zYVw · 2026-04-28

**Summary Of Contributions:**

This paper surveys the growing literature on flow matching (FM) for reinforcement learning. The authors set up a two-axis taxonomy: (i) what distribution is modeled (policies, critics, dynamics, preferences) and (ii) where the RL signal enters (weighted regression at training time, online gradients, GRPO-style updates). Sections 4–6 cover representative methods across offline RL (QIPO/EFM, FlowQ, FQL, ReFORM), online RL (ReinFlow, FMPG, PolicyFlow, the GRPO family, ORW-CFM-W2/ADRPO), and non-policy applications (floq for critics, CompFlow for dynamics, PFM for preferences). Sections 7–9 discuss emerging frontiers, compare FM against Gaussian/diffusion baselines, and list open problems.


**Strengths.** There is no existing survey dedicated to FM in RL, so the paper fills a real gap at the right time. The high-level framing — "offline FM-RL = weighted regression toward μexp⁡(βQ)\mu \exp(\beta Q)
μexp(βQ); online FM-RL = how to deal with implicit likelihoods from ODE-defined policies" — is genuinely helpful and connects methods that otherwise look quite different. Section 2 + Figure 1 give a clean, self-contained introduction to flow matching that works as a standalone primer. The algorithm boxes (Alg. 3, 4, 9 in particular) hit the right level of detail for a survey reader. Section 8's side-by-side comparison of Gaussian / diffusion / flow policies is a useful reference on its own.


**Weaknesses.** The two-axis taxonomy is stated in Section 3 but not actually used to organize Sections 4–6, which follow a simpler offline / online / beyond split. Section 7 lists papers one by one without much synthesis. Some technical descriptions are imprecise in ways that could mislead a reader who has not read the original paper. There is also quite a bit of overlap between Sections 2.4 and 3.2. No empirical comparison table is provided, which is a missed opportunity.

**Audience:**

Yes

**Audience Explanation:**

Flow matching has gone from a niche generative modeling technique to a widely-used policy class for offline RL, robotics, and alignment in a very short time, and the literature is scattered across different communities. Anyone entering this area or following it from a neighboring one (diffusion RL, RLHF, continuous normalizing flows) currently has no single reference to consult. A well-executed survey of this space would be cited regularly as a starting point, and TMLR is a natural home for it.

**Broader Impact Concerns:**

The paper includes a Broader Impact Statement that covers the dual-use risks of better policy learning for autonomous systems and alignment. This is adequate for a survey paper. No further concerns.

**Claims And Evidence:**

Yes

**Claims Explanation:**

The FM background (Section 2.3), the energy-weighted derivation (Sections 4.1–4.2), and the three-family split for online FM-RL (Section 5) match what the underlying papers say. There are a few local inaccuracies and some overstated entries in Table 3 (detailed under Requested Changes), but nothing that breaks the overall narrative. The taxonomy itself is reasonable, though the body of the paper does not follow through on the two-axis organization as cleanly as Section 3 promises.

**Requested Changes:**

1. The two-axis taxonomy does not match the paper's actual structure. Section 3 introduces two "orthogonal design axes" (what is modeled × where the RL signal enters), but Sections 4–6 are organized along just one axis: offline / online / beyond policies. Figure 3 also follows this one-axis layout. I think the fix is straightforward: either add a 2D summary table that genuinely crosses the two axes (rows = {actions, critics, dynamics, preferences}, columns = {weighted regression, ODE-backprop, GRPO, …}) or tone down the "two orthogonal axes" claim in Section 3 to match what the paper actually does.

2. Some method descriptions are not precise enough for a reader to trust. A few examples:

Section 4.2 (QIPO/EFM): The jump from the general guided-flow problem (Eq. 16) to the weighted conditional loss (Eq. 17) skips the key argument for why this works — i.e., why weighting by terminal-sample energies under the unguided conditional path recovers the guided marginal. This should be stated at least informally, or the reader should be pointed to the specific result in Zhang et al. 2025b.

*Section 4.5 (ReFORM):* The reflected ODE notation dx=v dt+dLtdx = v\,dt + dL_t
dx=vdt+dLt​ and the actual implementation ("projected/reflected Euler step") are not the same thing mathematically. The paper should be clearer about which one is the theoretical claim and which is the practical algorithm.

3. Add a benchmark comparison table. A survey of applied methods without any quantitative summary is harder to use in practice. Even a table that collects numbers reported by each method on standard benchmarks (D4RL locomotion/antmaze, Meta-World, GenEval or image-reward for the GRPO family) — with a note that these are not head-to-head reproductions — would be very helpful for readers trying to figure out which method to try.

4. Section 7 needs more synthesis. Right now each subsection reads like a list: paper name, one sentence, next paper. The reader gets names but not a picture of which ideas are converging, which tradeoffs are understood, or where the real gaps are. I'd suggest grouping papers into 2–3 themes and writing a short comparison within each, rather than going paper by paper.

5. Cut the overlap between Sections 2.4 and 3.2. Both sections explain the training-time vs. sampling-time distinction with overlapping examples (QIPO, FMPG, GRPO). Section 2.4 already does most of the work; Section 3.2 could be shortened to a paragraph that points forward to Sections 4–5.

---

> ### Author Response · Authors · 2026-05-11
>
> We sincerely thank the reviewer for the detailed helpful feedback. We have made all 5 requested changes in the revised manuscript.
>
> > (1) Two-axis taxonomy does not match the paper's structure, either add a 2D summary table or tone down the claim.
>
> We have added a 2D summary table (Table 1 in revised Section 3) that explicitly crosses both axes. Rows represent what the flow models (actions, critics, dynamics, preferences) and columns represent how the RL signal enters (weighted regression, ODE/pathwise gradients, GRPO, other). Empty cells mark under-explored combinations. An accompanying paragraph explains how Table 1 complements the offline/online/beyond narrative in Sections 4 to 6, and highlights gaps visible in the 2D view. For example, GRPO-style updates have been applied almost exclusively to action and preference flows, while weighted-regression critics and dynamics flows are still rare. Several methods occupy hybrid cells (for example, FQL combines a behavior-cloning flow with a separate one-step actor head).
>
> > (2a) QIPO/EFM (Section 4.2). The jump from Eq. 16 to Eq. 17 skips the key argument for why this works.
>
> We have added a "Why this works" paragraph between Equations 16 and 17. It states the importance-reweighting argument informally. Since $q_1(x_1) \propto p_1(x_1)\exp(-E(x_1))$, samples from $q_1$ can be obtained by reweighting $p_1$ samples with $w(x_1) \propto \exp(-E(x_1))$. Zhang et al. (2025b) extend this to intermediate times, showing that the conditional FM loss with terminal weights recovers the guided marginal velocity field without evaluating the intermediate energy $E_t(x)$. We point readers to Theorems 4.1 and 4.3 in Zhang et al. (2025b) for the formal proof.
>
> > (2b) ReFORM (Section 4.5). The reflected ODE notation and the actual implementation are not the same thing mathematically.
>
> We now clearly separate "Theoretical construction" (the reflected ODE with local-time process $L_t$ that formally guarantees boundary containment) from "Practical algorithm" (the projected/reflected Euler step that is a numerical surrogate preserving the support-containment invariant without exactly reproducing the continuous dynamics). We point readers to Eq. 9, Eq. 12, and Theorem 1 in Zhang et al. (2026).
>
> > (3) Add a benchmark comparison table.
>
> We have added three tables in the new Section 10 (Comparative Analysis). Table 5 consolidates offline RL results across D4RL Gym-Loco (9-task average), AntMaze (6-task average), and OGBench (50-task average) for Gaussian, diffusion, and flow-matching methods, reporting normalized return, NFE, and per-action generation time. Each table includes a "Reading Table" paragraph that synthesizes the key trends. For example, on AntMaze, FQL reaches 84 with a single-step actor while IDQL reaches 79.1 but requires 128 rejection samples per action, making the latency gap substantial in deployment. On OGBench, FQL achieves 43.8 versus 31.0 for the strongest Gaussian baseline (ReBRAC), widening the advantage of one-step distilled flow policies on benchmarks designed to expose the limits of unimodal policies. Table 6 covers online RL on MuJoCo Playground, DMC, and OpenAI Gym. Table 7 covers the GRPO family on text-to-image alignment (GenEval, OCR Accuracy, PickScore). All captions note these are not head-to-head reproductions. We also added a dedicated Section 9 (Benchmarks and Evaluation Protocols) covering benchmark families, evaluation metrics beyond return (NFE, wall-clock time), and the three axes along which fair comparisons should be made.
>
> > (4) "Section 7 needs more synthesis. Each subsection reads like a list."
>
> We reorganized Section 7 into thematic clusters with explicit tradeoff comparisons. Section 7.1 identifies shared design logic across discrete (CTMC), combinatorial (latent-spherical), and Riemannian (geodesic) extensions, noting that all three encode geometry in the transport process rather than via post-hoc projection. Section 7.2 organizes architectural innovations along a spectrum from backbone redesign (full expressiveness but nontrivial architecture changes) to residual correction (simplest to deploy but cannot move far from the prior) to single-step collapse (fastest inference but quality depends on low target variance). Section 7.3 synthesizes distributional flow critics (floq, DFC, FlowCritic, EVOR) around the tension between ODE integration cost and each method's solution. Section 7.4 frames intermediate credit assignment as the unifying problem behind alignment and trajectory-level works. A closing paragraph notes cross-cutting connections and remaining gaps.
>
> > (5) Cut the overlap between Sections 2.4 and 3.2.
>
> Section 3.2 is now a single paragraph that names each integration category, references Section 2.4 and points forward to Sections 4 and 5. This eliminates the redundancy while preserving parallel structure with Section 3.1.
>
> We are grateful to the reviewer for the genuinely helpful and constructive feedback.

---

> > ### Comment · Reviewer_zYVw · 2026-05-27
> >
> > I want to thank the authors for the detailed revision. I believe the manuscript is now in a good shape and ready for publication.

---

### Decision · Action_Editor_vn1B · 2026-06-02

**Recommendation:** Accept as is

**Audience:**

Yes

**Audience Explanation:**

Reviewers believe that this is a timely survey and is of great importance for the community.

**Claims And Evidence:**

Yes

**Claims Explanation:**

The submission provides a technically grounded synthesis of the flow-matching-in-reinforcement-learning literature. All reviewers are positive with the survey.